# A Measure-Theoretic Approach to Kernel Conditional Mean Embeddings

**Junhyung Park**
MPI for Intelligent Systems, Tübingen
`junhyung.park@tuebingen.mpg.de`

**Krikamol Muandet**
MPI for Intelligent Systems, Tübingen
`krikamol@tuebingen.mpg.de`

## Abstract

We present an operator-free, measure-theoretic approach to the conditional mean embedding (CME) as a random variable taking values in a reproducing kernel Hilbert space. While the kernel mean embedding of unconditional distributions has been defined rigorously, the existing operator-based approach of the conditional version depends on stringent assumptions that hinder its analysis. We overcome this limitation via a measure-theoretic treatment of CMEs. We derive a natural regression interpretation to obtain empirical estimates, and provide a thorough theoretical analysis thereof, including universal consistency. As natural by-products, we obtain the conditional analogues of the maximum mean discrepancy and Hilbert-Schmidt independence criterion, and demonstrate their behaviour via simulations.

## 1 Introduction

The idea of embedding probability distributions into a reproducing kernel Hilbert space (RKHS), a space associated to a positive definite kernel, has received a lot of attention in the past decades [1, 45], and has found a wealth of successful applications, such as independence testing [20], two-sample testing [21], learning on distributions [33, 30, 55], goodness-of-fit testing [8, 29] and probabilistic programming [41, 44], among others – see review [34]. It extends the idea of kernelising linear methods by embedding data points into high- (and often infinite-)dimensional RKHSs, which has been applied, for example, in ridge regression, spectral clustering, support vector machines and principal component analysis among others [40, 25, 52].

Conditional distributions can also be embedded into RKHSs in a similar manner [49],[34, Chapter 4]. Compared to unconditional distributions, conditional distributions can represent more complicated relations between random variables, and so conditional mean embeddings (CMEs) have the potential to unlock the arsenal of kernel mean embeddings to a wider setting. Indeed, CMEs have been applied successfully to dynamical systems [46], inference on graphical models [48], probabilistic inference via kernel sum and product rules [49], reinforcement learning [23, 35], kernelising the Bayes rule and applying it to nonparametric state-space models [17] and causal inference [32] to name a few.

Despite such progress, the current prevalent definition of the CME based on composing cross-covariance operators [46] relies on some stringent assumptions, which are often violated and hinder its analysis. Klebanov et al. [27] recently attempted to clarify and weaken some of these assumptions, but strong and hard-to-verify conditions still persist. Grünewälder et al. [22] provided a regression interpretation, but here, only the existence of the CME is shown, without an explicit expression. The main contribution of this paper is to remove these stringent assumptions using a novel measure-theoretic approach to the CME. This approach requires drastically weaker assumptions, and comes in an explicit expression. We believe this will enable a more principled analysis of its theoretical properties, and open doors to new application areas. We derive an empirical estimate based on vector-valued regression along with in-depth theoretical analysis, including universal consistency. In particular, we relax the assumption of [22] to allow for infinite-dimensional RKHSs.

As natural by-products, we obtain quantities that are extensions of the maximum mean discrepancy (MMD) and the Hilbert-Schmidt independence criterion (HSIC) to the conditional setting, which we call the *maximum conditional mean discrepancy* (MCMD) and the *Hilbert-Schmidt conditional independence criterion* (HSCIC). We demonstrate their properties through simulation experiments.

All proofs can be found in Appendix C.

## 2 Preliminaries

We take $(\Omega, \mathcal{F}, P)$ as the underlying probability space. Let $(\mathcal{X}, \mathfrak{X})$, $(\mathcal{Y}, \mathfrak{Y})$ and $(\mathcal{Z}, \mathfrak{Z})$ be separable measurable spaces, and let $X : \Omega \to \mathcal{X}$, $Y : \Omega \to \mathcal{Y}$ and $Z : \Omega \to \mathcal{Z}$ be random variables with distributions $P_X$, $P_Y$ and $P_Z$. We will use $Z$ as the conditioning variable throughout.

### 2.1 Positive definite kernels and RKHS embeddings

Let $\mathcal{H}_\mathcal{X}$ be a vector space of $\mathcal{X} \to \mathbb{R}$ functions, endowed with a Hilbert space structure via an inner product $\langle \cdot, \cdot \rangle_{\mathcal{H}_\mathcal{X}}$. A symmetric function $k_\mathcal{X} : \mathcal{X} \times \mathcal{X} \to \mathbb{R}$ is a *reproducing kernel* of $\mathcal{H}_\mathcal{X}$ if and only if: 1. $\forall x \in \mathcal{X}$, $k_\mathcal{X}(x, \cdot) \in \mathcal{H}_\mathcal{X}$; 2. $\forall x \in \mathcal{X}$ and $\forall f \in \mathcal{H}_\mathcal{X}$, $f(x) = \langle f, k_\mathcal{X}(x, \cdot) \rangle_{\mathcal{H}_\mathcal{X}}$. A space $\mathcal{H}_\mathcal{X}$ which possesses a reproducing kernel is called a *reproducing kernel Hilbert space* (RKHS) [1]. Throughout this paper, we assume that all RKHSs are *separable*. This is not a restrictive assumption, since it is satisfied if, for example, $k_\mathcal{X}$ is a continuous kernel [52, p.130, Lemma 4.33] (for further details, please see [36]). Given a distribution $P_X$ on $\mathcal{X}$, assuming the integrability condition

$$\int_\mathcal{X} \sqrt{k_\mathcal{X}(x, x)} dP_X(x) < \infty, \tag{1}$$

we define the *kernel mean embedding* $\mu_{P_X}(\cdot) = \int_\mathcal{X} k_\mathcal{X}(x, \cdot) dP_X(x)$ of $P_X$, where the integral is a *Bochner integral* [12, p.15, Def. 35]. We will later show a conditional analogue of the following lemma (for completeness, a proof is provided in Appendix C).

**Lemma 2.1** ([45]). *For each $f \in \mathcal{H}_\mathcal{X}$, $\int_\mathcal{X} f(x) dP_X(x) = \langle f, \mu_{P_X} \rangle_{\mathcal{H}_\mathcal{X}}$.*

Next, suppose $\mathcal{H}_\mathcal{Y}$ is an RKHS of functions on $\mathcal{Y}$ with kernel $k_\mathcal{Y}$, and consider the *tensor product RKHS $\mathcal{H}_\mathcal{X} \otimes \mathcal{H}_\mathcal{Y}$* (see [58, pp.47-48] for a definition of tensor product Hilbert spaces).

**Theorem 2.2** ([1, p.31, Theorem 13]). *The tensor product $\mathcal{H}_\mathcal{X} \otimes \mathcal{H}_\mathcal{Y}$ is generated by the functions $f \otimes g : \mathcal{X} \times \mathcal{Y} \to \mathbb{R}$, with $f \in \mathcal{H}_\mathcal{X}$ and $g \in \mathcal{H}_\mathcal{Y}$ defined by $(f \otimes g)(x, y) = f(x)g(y)$. Moreover, $\mathcal{H}_\mathcal{X} \otimes \mathcal{H}_\mathcal{Y}$ is an RKHS of functions on $\mathcal{X} \times \mathcal{Y}$ with kernel $(k_\mathcal{X} \otimes k_\mathcal{Y})((x_1, y_1), (x_2, y_2)) = k_\mathcal{X}(x_1, x_2)k_\mathcal{Y}(y_1, y_2)$.*

Now let us impose a slightly stronger integrability condition:

$$\mathbb{E}_X[k_\mathcal{X}(X, X)] < \infty, \quad \mathbb{E}_Y[k_\mathcal{Y}(Y, Y)] < \infty. \tag{2}$$

This ensures that $k_\mathcal{X}(X, \cdot) \otimes k_\mathcal{Y}(Y, \cdot)$ is Bochner $P_{XY}$-integrable, and so $\mu_{P_{XY}} := \mathbb{E}_{XY}[k_\mathcal{X}(X, \cdot) \otimes k_\mathcal{Y}(Y, \cdot)] \in \mathcal{H}_\mathcal{X} \otimes \mathcal{H}_\mathcal{Y}$. The next lemma is analogous to Lemma 2.1:

**Lemma 2.3** ([15, Theorem 1]). *For $f \in \mathcal{H}_\mathcal{X}$, $g \in \mathcal{H}_\mathcal{Y}$, $\langle f \otimes g, \mu_{P_{XY}} \rangle_{\mathcal{H}_\mathcal{X} \otimes \mathcal{H}_\mathcal{Y}} = \mathbb{E}_{XY}[f(X)g(Y)]$.*

As a consequence, for any pair $f \in \mathcal{H}_\mathcal{X}$ and $g \in \mathcal{H}_\mathcal{Y}$, we have $\langle f \otimes g, \mu_{P_{XY}} - \mu_{P_X} \otimes \mu_{P_Y} \rangle_{\mathcal{H}_\mathcal{X} \otimes \mathcal{H}_\mathcal{Y}} = \mathrm{Cov}_{XY}[f(X), g(Y)]$. There exists an isometric isomorphism $T : \mathcal{H}_\mathcal{X} \otimes \mathcal{H}_\mathcal{Y} \to \mathrm{HS}(\mathcal{H}_\mathcal{X}, \mathcal{H}_\mathcal{Y})$, where $\mathrm{HS}(\mathcal{H}_\mathcal{X}, \mathcal{H}_\mathcal{Y})$ is the space of Hilbert-Schmidt operators from $\mathcal{H}_\mathcal{X}$ to $\mathcal{H}_\mathcal{Y}$ (Lemma C.1). The (centred) *cross-covariance operator* is defined as $\mathcal{C}_{YX} := T(\mu_{P_{XY}} - \mu_{P_X} \otimes \mu_{P_Y})$ [15, Theorem 1]. The object $T(\mu_{P_{XY}})$ is referred to as the *uncentred cross-covariance operator* in the literature [47, Section 3.2].

The notion of *characteristic kernels* is essential, since it tells us that the associated RKHSs are rich enough to enable us to distinguish different distributions from their embeddings.

**Definition 2.4** ([16]). A positive definite kernel $k_\mathcal{X}$ is *characteristic* to a set $\mathcal{P}$ of probability measures defined on $\mathcal{X}$ if the map $\mathcal{P} \to \mathcal{H}_\mathcal{X} : P_X \mapsto \mu_{P_X}$ is injective.

Sriperumbudur et al. [50] discusses various characterisations of characteristic kernels and show that the well-known Gaussian and Laplacian kernels are characteristic. We then have a metric on $\mathcal{P}$ via $\|\mu_{P_X} - \mu_{P_{X'}}\|_{\mathcal{H}_\mathcal{X}}$ for $P_X, P_{X'} \in \mathcal{P}$, which is the definition of the MMD [19]. Furthermore, the HSIC is defined as the Hilbert-Schmidt norm of $\mathcal{C}_{YX}$, or equivalently, $\|\mu_{P_{XY}} - \mu_{P_X} \otimes \mu_{P_Y}\|_{\mathcal{H}_\mathcal{X} \otimes \mathcal{H}_\mathcal{Y}}$ [18]. If $k_\mathcal{X} \otimes k_\mathcal{Y}$ is characteristic, then HSIC = 0 if and only if $X \perp\!\!\!\perp Y$.

## 2.2 Conditioning

We briefly review the concept of conditioning in measure-theoretic probability theory, with Banach space-valued random variables. We consider a sub-$\sigma$-algebra $\mathcal{E}$ of $\mathcal{F}$ and a Banach space $\mathcal{H}$.

**Definition 2.5** (Conditional Expectation, [12, p.45, Definition 38]). Suppose $H$ is a Bochner $P$-integrable, $\mathcal{H}$-valued random variable. Then the *conditional expectation* of $H$ given $\mathcal{E}$ is any $\mathcal{E}$-measurable, Bochner $P$-integrable, $\mathcal{H}$-valued random variable $H'$ such that $\int_A H \, dP = \int_A H' \, dP$ $\forall A \in \mathcal{E}$. Any $H'$ satisfying this condition is a *version* of $\mathbb{E}[H \mid \mathcal{E}]$. We write $\mathbb{E}[H \mid Z]$ to mean $\mathbb{E}[H \mid \sigma(Z)]$, where $\sigma(Z)$ is the sub-$\sigma$-algebra of $\mathcal{F}$ generated by the random variable $Z$.

The (almost sure) uniqueness of the conditional expectation is shown in [12, p.44, Proposition 37], and the existence in [12, pp.45-46, Theorems 39 and 50].

**Definition 2.6** ([9, p.149]). The *conditional probability* of $A \in \mathcal{F}$ given $\mathcal{E}$ is $P(A \mid \mathcal{E}) = \mathbb{E}[\mathbf{1}_A \mid \mathcal{E}]$.

Note that, in the unconditional case, the expectation is defined as the integral with respect to the measure, but in the conditional case, the expectation is defined *first*, and the measure is *defined* as the expectation of the indicator function. For this definition to be useful, we require an additional property, called *regular version*. We first define the *transition probability kernel*[1].

**Definition 2.7** ([9, p.37,40]). Let $(\Omega_i, \mathcal{F}_i)$, $i = 1, 2$ be measurable spaces. A mapping $K : \Omega_1 \times \mathcal{F}_2 \to [0, \infty]$ is a *transition kernel* from $(\Omega_1, \mathcal{F}_1)$ to $(\Omega_2, \mathcal{F}_2)$ if (i) $\forall B \in \mathcal{F}_2$, $\omega \mapsto K(\omega, B)$ is $\mathcal{F}_1$-measurable; (ii) $\forall \omega \in \Omega_1$, $B \mapsto K(\omega, B)$ is a measure on $(\Omega_2, \mathcal{F}_2)$. If $K(\omega, \Omega_2) = 1$ $\forall \omega \in \Omega_1$, $K$ is said to be a *transition probability kernel*.

**Definition 2.8** ([9, p.150, Definition 2.4]). For each $A \in \mathcal{F}$, let $Q(A)$ be a version of $P(A|\mathcal{E}) = \mathbb{E}[\mathbf{1}_A|\mathcal{E}]$. Then $Q : (\omega, A) \mapsto Q_\omega(A)$ is said to be a *regular version* of the conditional probability measure $P(\cdot \mid \mathcal{E})$ if $Q$ is a transition probability kernel from $(\Omega, \mathcal{E})$ to $(\Omega, \mathcal{F})$.

The following theorem, proved in Appendix C, is the reason why a regular version is important. It means that, roughly speaking, the conditional expectation is indeed obtained by integration with respect to the conditional measure.

**Theorem 2.9** (Adapted from [9, p.150, Proposition 2.5]). *Suppose that $P(\cdot \mid \mathcal{E})$ admits a regular version $Q$. Then $QH : \Omega \to \mathcal{H}$ with $\omega \mapsto Q_\omega H = \int_\Omega H(\omega') Q_\omega(d\omega')$ is a version of $\mathbb{E}[H \mid \mathcal{E}]$ for every Bochner $P$-integrable $H$.*

## 2.3 Vector-valued RKHS regression

In this subsection, we introduce the theory of vector-valued RKHS regression, based on operator-valued kernels. Let $\mathcal{H}$ be a Hilbert space, which will be the output space of regression.

**Definition 2.10** ([6, Definition 1]). An *$\mathcal{H}$-valued RKHS* on $\mathcal{Z}$ is a Hilbert space $\mathcal{G}$ such that 1. the elements of $\mathcal{G}$ are functions $\mathcal{Z} \to \mathcal{H}$; 2. $\forall z \in \mathcal{Z}$, $\exists C_z > 0$ such that $\|F(z)\|_{\mathcal{H}} \leq C_z \|F\|_{\mathcal{G}}$ $\forall F \in \mathcal{G}$.

Next, we let $\mathcal{L}(\mathcal{H})$ denote the Banach space of bounded linear operators from $\mathcal{H}$ into itself.

**Definition 2.11** ([6, Definition 2]). A *$\mathcal{H}$-kernel of positive type* on $\mathcal{Z} \times \mathcal{Z}$ is a map $\Gamma : \mathcal{Z} \times \mathcal{Z} \to \mathcal{L}(\mathcal{H})$ such that $\forall N \in \mathbb{N}$, $\forall z_1, ..., z_N \in \mathcal{Z}$ and $\forall c_1, ..., c_N \in \mathbb{R}$, $\sum_{i,j=1}^{N} c_i c_j \langle \Gamma(z_j, z_i) h, h \rangle_{\mathcal{H}} \geq 0$ $\forall h \in \mathcal{H}$.

Analogously to the scalar case, it can be shown that any $\mathcal{H}$-valued RKHS $\mathcal{G}$ possesses a *reproducing kernel*, which is an $\mathcal{H}$-kernel of positive type $\Gamma$ satisfying, for any $z, z' \in \mathcal{Z}$, $h, h' \in \mathcal{H}$ and $F \in \mathcal{G}$, $\langle F(z), h \rangle_{\mathcal{H}} = \langle F, \Gamma(\cdot, z) h \rangle_{\mathcal{G}}$ and $\langle h, \Gamma(z, z')(h') \rangle_{\mathcal{H}} = \langle \Gamma(\cdot, z)(h), \Gamma(\cdot, z')(h') \rangle_{\mathcal{G}}$.

Now suppose we want to perform regression with input space $\mathcal{Z}$ and output space $\mathcal{H}$, by minimising

$$\frac{1}{n} \sum_{j=1}^{n} \|h_j - F(z_j)\|_{\mathcal{H}}^2 + \lambda \|F\|_{\mathcal{G}}^2, \tag{3}$$

where $\lambda > 0$ is a regularisation parameter and $\{(z_j, h_j) : j = 1, ..., n\} \subseteq \mathcal{Z} \times \mathcal{H}$. There is a corresponding representer theorem (here, $\delta_{jl}$ is the Kronecker delta):

**Theorem 2.12** ([31, Theorem 4.1]). *If $\hat{F}$ minimises (3) in $\mathcal{G}$, it is unique and has the form $\hat{F} = \sum_{j=1}^{n} \Gamma(\cdot, z_j)(u_j)$ where the coefficients $\{u_j : j = 1, ..., n\} \subseteq \mathcal{H}$ are the unique solution of the linear equations $\sum_{l=1}^{n} (\Gamma(z_j, z_l) + n\lambda \delta_{jl})(u_l) = h_j, j = 1, ..., n$.*

# 3 Conditional mean embedding

We are now ready to introduce a formal definition of the conditional mean embedding of $X$ given $Z$.

**Definition 3.1.** Assuming $X$ satisfies the integrability condition (1), we define the *conditional mean embedding* of $X$ given $Z$ as $\mu_{P_{X|Z}} := \mathbb{E}_{X|Z}[k_{\mathcal{X}}(X, \cdot) \mid Z]$.

This is a direct extension of the unconditional kernel mean embedding, $\mu_{P_X} = \mathbb{E}_X[k_{\mathcal{X}}(X, \cdot)]$, but instead of being a fixed element in $\mathcal{H}_{\mathcal{X}}$, $\mu_{P_{X|Z}}$ is a $Z$-measurable random variable taking values in $\mathcal{H}_{\mathcal{X}}$ (see Definition 2.5). Also, for any function $f : \mathcal{X} \to \mathbb{R}$, $\mathbb{E}_{X|Z}[f(X) \mid Z]$ is a real-valued $Z$-measurable random variable. The following lemma is analogous to Lemma 2.1.

**Lemma 3.2.** *For any $f \in \mathcal{H}_{\mathcal{X}}$, $\mathbb{E}_{X|Z}[f(X) \mid Z] = \langle f, \mu_{P_{X|Z}} \rangle_{\mathcal{H}_{\mathcal{X}}}$ almost surely.*

Next, assuming $X$ and $Y$ satisfy (2), we define $\mu_{P_{XY|Z}} := \mathbb{E}_{XY|Z}[k_{\mathcal{X}}(X, \cdot) \otimes k_{\mathcal{Y}}(Y, \cdot) \mid Z]$, a $Z$-measurable, $\mathcal{H}_{\mathcal{X}} \otimes \mathcal{H}_{\mathcal{Y}}$-valued random variable. We have the following analogy of Lemma 2.3:

**Lemma 3.3.** *For any pair $f \in \mathcal{H}_{\mathcal{X}}$ and $g \in \mathcal{H}_{\mathcal{Y}}$, $\mathbb{E}_{XY|Z}[f(X)g(Y) \mid Z] = \langle f \otimes g, \mu_{P_{XY|Z}} \rangle_{\mathcal{H}_{\mathcal{X}} \otimes \mathcal{H}_{\mathcal{Y}}}$ almost surely.*

By Lemmas 3.2 and 3.3, for any pair $f \in \mathcal{H}_{\mathcal{X}}$ and $g \in \mathcal{H}_{\mathcal{Y}}$,

$$\langle f \otimes g, \mu_{P_{XY|Z}} - \mu_{P_{X|Z}} \otimes \mu_{P_{Y|Z}} \rangle_{\mathcal{H}_{\mathcal{X}} \otimes \mathcal{H}_{\mathcal{Y}}} = \mathrm{Cov}_{XY|Z}(f(X), g(Y) \mid Z)$$
$$= \mathbb{E}_{XY|Z}[f(X)g(Y) \mid Z] - \mathbb{E}_{X|Z}[f(X) \mid Z]\mathbb{E}_{Y|Z}[g(Y) \mid Z]$$

almost surely. Hence, we define the *conditional cross-covariance operator* as $\mathcal{C}_{YX|Z} := T(\mu_{P_{XY|Z}} - \mu_{P_{X|Z}} \otimes \mu_{P_{Y|Z}})$ (see Section 2.1 for the definition of $T$).

## 3.1 Comparison with existing definitions

As previously mentioned, the idea of CMEs and conditional cross-covariance operators is not a novel one, yet our development of the theory above differs significantly from the existing works. In this subsection, we review the previous approaches and compare them to ours.

The prevalent definition of CMEs in the literature is the following. We first need to endow the conditioning space $\mathcal{Z}$ with a scalar kernel, say $k_{\mathcal{Z}} : \mathcal{Z} \times \mathcal{Z} \to \mathbb{R}$, with corresponding RKHS $\mathcal{H}_{\mathcal{Z}}$.

**Definition 3.4** ([46, Definition 3]). The conditional mean embedding of the conditional distribution $P(X \mid Z)$ is the operator $\mathcal{U}_{X|Z} : \mathcal{H}_{\mathcal{Z}} \to \mathcal{H}_{\mathcal{X}}$ defined by $\mathcal{U}_{X|Z} = \mathcal{C}_{XZ}\mathcal{C}_{ZZ}^{-1}$, where $\mathcal{C}_{XZ}$ and $\mathcal{C}_{ZZ}$ are unconditional (cross-)covariance operators as defined in Section 2.1.

As noted by [46], the motivation for this comes from [15, Theorem 2], which states that for any $f \in \mathcal{H}_{\mathcal{X}}$, if $\mathbb{E}_{X|Z}[f(X) \mid Z = \cdot] \in \mathcal{H}_{\mathcal{Z}}$, then $\mathcal{C}_{ZZ}\mathbb{E}_{X|Z}[f(X) \mid Z = \cdot] = \mathcal{C}_{ZX}f$. This relation can be used to prove the following theorem, which is analogous to Lemma 3.2.

**Theorem 3.5** ([46, Theorem 4]). *For $f \in \mathcal{H}_{\mathcal{X}}$, assuming $\mathbb{E}_{X|Z}[f(X) \mid Z = \cdot] \in \mathcal{H}_{\mathcal{Z}}$, $\mathcal{U}_{X|Z}$ satisfies: 1. $\mu_{X|z} := \mathbb{E}_{X|z}[k_{\mathcal{X}}(X, \cdot) \mid Z = z] = \mathcal{U}_{X|Z}k_{\mathcal{Z}}(z, \cdot)$; 2. $\mathbb{E}_{X|z}[f(X) \mid Z = z] = \langle f, \mu_{X|z} \rangle_{\mathcal{H}_{\mathcal{X}}}$.*

Now we highlight the key differences between this approach and ours. Firstly, this approach requires the endowment of a kernel $k_{\mathcal{Z}}$ on the conditioning space $\mathcal{Z}$, and defines the CME as an *operator* from $\mathcal{H}_{\mathcal{Z}}$ to $\mathcal{H}_{\mathcal{X}}$. By contrast, Definition 3.1 did not consider any kernel or function on $\mathcal{Z}$, and defined the CME as a *Bochner conditional expectation* given $\sigma(Z)$. We argue that it is more natural not to endow the *conditioning space* with a kernel before the estimation stage. Secondly, the operator-based approach assumes that $\mathbb{E}_{X|Z}[f(X)|Z = \cdot]$, as a function in $z$, lives in $\mathcal{H}_{\mathcal{Z}}$. This is a severe restriction; it is stated in [46] that this assumption, while true for finite domains with characteristic kernels, is not necessarily true for continuous domains, and [17] gives a simple counterexample using the Gaussian kernel. Lastly, it also assumes that $\mathcal{C}_{ZZ}^{-1}$ exists, which is another unrealistic assumption. [17] mentions that this assumption is too strong in many situations, and gives a counterexample using the Gaussian kernel. The most common remedy is to resort to the regularised version $\mathcal{C}_{XZ}(\mathcal{C}_{ZZ} + \lambda I)^{-1}$ and treat it as an approximation of $\mathcal{U}_{X|Z}$. These assumptions have been clarified and slightly weakened in [27], but strong and hard-to-verify conditions persist. In contrast, Definition 3.1 extend the notions of kernel mean embedding, expectation operator and cross-covariance operator to the conditional setting simply by using the formal definition of conditional expectations (Definition 2.5), and the subsequent result in Lemma 3.2, analogous to [46, Theorem 4], does not rely on any assumptions.

A regression interpretation is given in [22], by showing the *existence*, for each $z \in \mathcal{Z}$, of $\mu(z) \in \mathcal{H}_{\mathcal{X}}$ that satisfies $\mathbb{E}[h(X) \mid Z = z] = \langle h, \mu(z) \rangle_{\mathcal{H}_{\mathcal{X}}}$. However, no explicit expression for $\mu(z)$ is provided. In contrast, our definition provides an explicit expression $\mu_{P_{X|Z}} = \mathbb{E}_{X|Z}[k_{\mathcal{X}}(X, \cdot) \mid Z]$.

In [15, Section A.2], the conditional cross-covariance operator is defined, but in a significantly different way. It is defined as $\Sigma_{YX|Z} := \mathcal{C}_{YX} - \mathcal{C}_{YZ}\tilde{\mathcal{C}}_{ZZ}^{-1}\mathcal{C}_{ZX}$, where $\tilde{\mathcal{C}}_{ZZ}^{-1}$ is the right inverse of $\mathcal{C}_{ZZ}$ on $(\text{Ker}\mathcal{C}_{ZZ})^{\perp}$. This has the property that, for all $f \in \mathcal{H}_{\mathcal{X}}$ and $g \in \mathcal{H}_{\mathcal{Y}}$, $\langle g, \Sigma_{YX|Z}f \rangle_{\mathcal{H}_{\mathcal{Y}}} = \mathbb{E}_Z[\text{Cov}_{XY|Z}(f(X), g(Y) \mid Z)]$. Note that this is different to our relation stated after Lemma 3.3; the conditional covariance is integrated out over $\mathcal{Z}$. In fact, this difference is explicitly noted by [46].

# 4 Empirical estimates

In this section, we discuss how we can obtain empirical estimates of $\mu_{P_{X|Z}} = \mathbb{E}_{X|Z}[k_{\mathcal{X}}(X, \cdot) \mid Z]$.

**Theorem 4.1.** *Denote the Borel $\sigma$-algebra of $\mathcal{H}_{\mathcal{X}}$ by $\mathcal{B}(\mathcal{H}_{\mathcal{X}})$. Then we can write $\mu_{P_{X|Z}} = F_{P_{X|Z}} \circ Z$, where $F_{P_{X|Z}} : \mathcal{Z} \to \mathcal{H}_{\mathcal{X}}$ is some deterministic function, measurable with respect to $\mathfrak{Z}$ and $\mathcal{B}(\mathcal{H}_{\mathcal{X}})$.*

Hence, estimating $\mu_{P_{X|Z}}$ boils down to estimating the function $F_{P_{X|Z}}$, which is exactly the setting for vector-valued regression (Section 2.3) with input space $\mathcal{Z}$ and output space $\mathcal{H}_{\mathcal{X}}$. In contrast to [22], where regression is motivated by applying the Riesz representation theorem conditioned on each value of $z \in \mathcal{Z}$, we derive the CME as an explicit function of $Z$, which we argue is a more principled way to motivate regression. Moreover, for continuous $Z$, the event $Z = z$ has measure 0, so it is not measure-theoretically rigorous to apply the Riesz representation theorem conditioned on $Z = z$.

The natural optimisation problem is to minimise the loss $\mathcal{E}_{X|Z}(F) := \mathbb{E}_Z[\|F_{P_{X|Z}}(Z) - F(Z)\|^2_{\mathcal{H}_{\mathcal{X}}}]$ among all $F \in \mathcal{G}_{\mathcal{X}\mathcal{Z}}$, where $\mathcal{G}_{\mathcal{X}\mathcal{Z}}$ is a vector-valued RKHS of functions $\mathcal{Z} \to \mathcal{H}_{\mathcal{X}}$. For simplicity, we endow $\mathcal{G}_{\mathcal{X}\mathcal{Z}}$ with a kernel $l_{\mathcal{X}\mathcal{Z}}(z, z') = k_{\mathcal{Z}}(z, z')\text{Id}$, where $k_{\mathcal{Z}}(\cdot, \cdot)$ is a scalar kernel on $\mathcal{Z}$.[2]

We cannot minimise $\mathcal{E}_{X|Z}$ directly, since we do not observe samples from $\mu_{P_{X|Z}}$, but only the pairs $(x_i, z_i)$ from $(X, Z)$. We bound this with a surrogate loss $\tilde{\mathcal{E}}_{X|Z}$ that has a sample-based version:

$$\mathcal{E}_{X|Z}(F) = \mathbb{E}_Z[\|\mathbb{E}_{X|Z}[k_{\mathcal{X}}(X, \cdot) - F(Z) \mid Z]\|^2_{\mathcal{H}_{\mathcal{X}}}] \leq \mathbb{E}_Z\mathbb{E}_{X|Z}[\|k_{\mathcal{X}}(X, \cdot) - F(Z)\|^2_{\mathcal{H}_{\mathcal{X}}} \mid Z]$$
$$= \mathbb{E}_{X,Z}[\|k_{\mathcal{X}}(X, \cdot) - F(Z)\|^2_{\mathcal{H}_{\mathcal{X}}}] =: \tilde{\mathcal{E}}_{X|Z}(F),$$

where we used generalised conditional Jensen's inequality (see Appendix A, or [38]). Section 4.1 discusses the meaning of this surrogate loss. We replace the surrogate population loss with a regularised empirical loss based on samples $\{(x_i, z_i)\}_{i=1}^n$ from the joint distribution $P_{XZ}$: $\hat{\mathcal{E}}_{X|Z,n,\lambda}(F) := \frac{1}{n}\sum_{i=1}^n \|k_{\mathcal{X}}(x_i, \cdot) - F(z_i)\|^2_{\mathcal{H}_{\mathcal{X}}} + \lambda\|F\|^2_{\mathcal{G}_{\mathcal{X}\mathcal{Z}}}$, where $\lambda > 0$ is a regularisation parameter. We see that this loss functional is exactly in the form of (3). Therefore, by Theorem 2.12, the minimiser $\hat{F}_{P_{X|Z},n,\lambda}$ of $\hat{\mathcal{E}}_{X|Z,n,\lambda}$ is $\hat{F}_{P_{X|Z},n,\lambda}(\cdot) = \mathbf{k}_Z^T(\cdot)\mathbf{f}$, where $\mathbf{k}_Z(\cdot) := (k_{\mathcal{Z}}(z_1, \cdot), ..., k_{\mathcal{Z}}(z_n, \cdot))^T$, $\mathbf{f} := (f_1, ..., f_n)^T$ and the coefficients $f_i \in \mathcal{H}_{\mathcal{X}}$ are the unique solutions of the linear equations $(\mathbf{K}_Z + n\lambda\mathbf{I})\mathbf{f} = \mathbf{k}_X$, where $[\mathbf{K}_Z]_{ij} := k_{\mathcal{Z}}(z_i, z_j)$, $\mathbf{k}_X := (k_{\mathcal{X}}(x_1, \cdot), ..., k_{\mathcal{X}}(x_n, \cdot))^T$ and $\mathbf{I}$ is the $n \times n$ identity matrix. Hence, the coefficients are $\mathbf{f} = \mathbf{W}\mathbf{k}_X$, where $\mathbf{W} = (\mathbf{K}_Z + n\lambda\mathbf{I})^{-1}$. Finally, substituting this into the expression for $\hat{F}_{P_{X|Z},n,\lambda}(\cdot)$, we have

$$\hat{F}_{P_{X|Z},n,\lambda}(\cdot) = \mathbf{k}_Z^T(\cdot)\mathbf{W}\mathbf{k}_X \in \mathcal{G}_{\mathcal{X}\mathcal{Z}}. \tag{4}$$

## 4.1 Surrogate loss, universality and consistency

In this subsection, we investigate the meaning and consequences of using the surrogate loss $\tilde{\mathcal{E}}_{X|Z}$ instead of the original $\mathcal{E}_{X|Z}$, as well as the universal consistency property of our learning algorithm.

Denote by $L^2(\mathcal{Z}, P_Z; \mathcal{H}_{\mathcal{X}})$ the Banach space of (equivalence classes of) measurable functions $F : \mathcal{Z} \to \mathcal{H}_{\mathcal{X}}$ such that $\|F(\cdot)\|^2_{\mathcal{H}_{\mathcal{X}}}$ is $P_Z$-integrable, with norm $\|F\|_2 = (\int_{\mathcal{Z}} \|F(z)\|^2_{\mathcal{H}_{\mathcal{X}}} dP_Z(z))^{\frac{1}{2}}$.

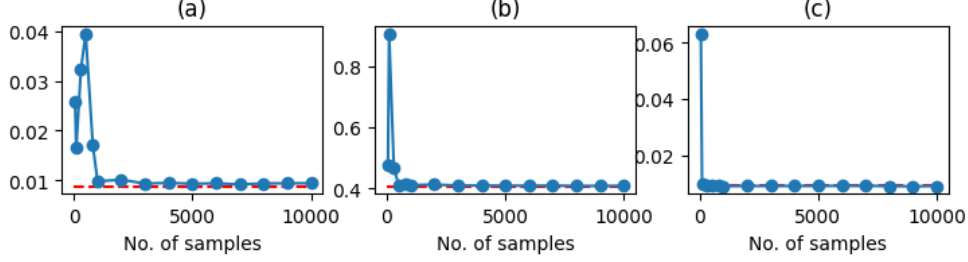

Figure 1: Solid blue and dashed red lines represent $\tilde{\mathcal{E}}_{X|Z}(\hat{F}_{P_{X|Z},n,\lambda_n})$ and $\tilde{\mathcal{E}}_{X|Z}(F_{P_{X|Z}})$ respectively.

We can note that the true function $F_{P_{X|Z}}$ belongs to $L^2(\mathcal{Z}, P_Z; \mathcal{H}_{\mathcal{X}})$, because Theorem 4.1 tells us that $F_{P_{X|Z}}$ is indeed measurable, and by Theorem A.2 and (2), $\int_{\mathcal{Z}} \|F_{P_{X|Z}}(z)\|^2_{\mathcal{H}_{\mathcal{X}}} dP_Z(z) = \mathbb{E}_Z[\|\mathbb{E}_{X|Z}[k_{\mathcal{X}}(X, \cdot) \mid Z]\|^2_{\mathcal{H}_{\mathcal{X}}}] \leq \mathbb{E}_Z[\mathbb{E}_{X|Z}[\|k_{\mathcal{X}}(X, \cdot)\|^2_{\mathcal{H}_{\mathcal{X}}} \mid Z]] = \mathbb{E}_X[\|k_{\mathcal{X}}(X, \cdot)\|^2_{\mathcal{H}_{\mathcal{X}}}] < \infty$. The true function $F_{P_{X|Z}}$ is the unique minimiser in $L^2(\mathcal{Z}, P_Z; \mathcal{H}_{\mathcal{X}})$ of both $\mathcal{E}_{X|Z}$ and $\tilde{\mathcal{E}}_{X|Z}$:

**Theorem 4.2.** $F_{P_{X|Z}}$ *minimises both* $\tilde{\mathcal{E}}_{X|Z}$ *and* $\mathcal{E}_{X|Z}$ *in* $L^2(\mathcal{Z}, P_Z; \mathcal{H}_{\mathcal{X}})$. *Moreover, it is almost surely equal to any other minimiser of the loss functionals.*

Note the difference in the statement of Theorem 4.2 from [22, Theorem 3.1], which only considers the minimisation of the loss functionals in $\mathcal{G}_{\mathcal{X}\mathcal{Z}}$, whereas we consider the larger space $L^2(\mathcal{Z}, P_Z; \mathcal{H}_{\mathcal{X}})$. Next, we discuss the concepts of *universal kernels* and *universal consistency*.

**Definition 4.3** ([7, Definition 2]). A kernel $l_{\mathcal{X}\mathcal{Z}} : \mathcal{Z} \times \mathcal{Z} \to \mathcal{L}(\mathcal{H}_{\mathcal{X}})$ with RKHS $\mathcal{G}_{\mathcal{X}\mathcal{Z}}$ is $\mathcal{C}_0$ if $\mathcal{G}_{\mathcal{X}\mathcal{Z}}$ is a subspace of $\mathcal{C}_0(\mathcal{Z}, \mathcal{H}_{\mathcal{X}})$, the space of continuous functions $\mathcal{Z} \to \mathcal{H}_{\mathcal{X}}$ vanishing at infinity. The kernel $l_{\mathcal{X}\mathcal{Z}}$ is $\mathcal{C}_0$-*universal* if is is $\mathcal{C}_0$ and $\mathcal{G}_{\mathcal{X}\mathcal{Z}}$ is dense in $L^2(\mathcal{Z}, P_Z; \mathcal{H}_{\mathcal{X}})$ for any measure $P_Z$ on $\mathcal{Z}$.

Carmeli et al. [7, Example 14] shows that $l_{\mathcal{X}\mathcal{Z}} = k_{\mathcal{Z}}(\cdot, \cdot)\text{Id}$ is $\mathcal{C}_0$-universal if $k_{\mathcal{Z}}$ is a universal scalar kernel, which in turn is guaranteed if $k_{\mathcal{Z}}$ is Gaussian or Laplacian, for example [51]. The consistency result with optimal rate $\mathcal{O}_p(\frac{\log n}{n})$ in [22, Corollaries 4.1, 4.2] is based on [5], and assumes, along with some distributional assumptions, that $\mathcal{H}_{\mathcal{X}}$ is finite-dimensional, which is not true for many common choices of $k_{\mathcal{X}}$ (see Appendix B for more details). In [46, Theorem 6], [48, Theorem 1] and [14, Theorem 1.3.2], consistency is also shown under various assumptions, with rates at best $\mathcal{O}_p(n^{-\frac{1}{4}})$. In Theorem 4.4, we prove universal consistency without any distributional assumptions, and in Theorem 4.5, we show that a convergence rate of $\mathcal{O}_p(n^{-1/4})$ can be achieved with a simple smoothness assumption that $F_{P_{X|Z}} \in \mathcal{G}_{\mathcal{X}\mathcal{Z}}$ (sometimes referred to as the *well-specified case*; see [55]). In particular, both results relax the finite-dimensionality assumption on $\mathcal{H}_{\mathcal{X}}$ of [22].

**Theorem 4.4.** *Suppose that* $k_{\mathcal{X}}$ *and* $k_{\mathcal{Z}}$ *are bounded kernels, i.e. there are* $B_{\mathcal{Z}}, B_{\mathcal{X}} > 0$ *with* $\sup_{z \in \mathcal{Z}} k_{\mathcal{Z}}(z, z) \leq B_{\mathcal{Z}}^2$, $\sup_{x \in \mathcal{X}} k_{\mathcal{X}}(x, x) \leq B_{\mathcal{X}}^2$, *and that the operator-valued kernel* $l_{\mathcal{X}\mathcal{Z}}$ *is* $\mathcal{C}_0$-*universal. Let the regularisation parameter* $\lambda_n$ *decay to 0 at a slower rate than* $\mathcal{O}(n^{-1/2})$. *Then the learning algorithm that yields* $\hat{F}_{P_{X|Z},n,\lambda_n}$ *is universally consistent, i.e. for any joint distribution* $P_{XZ}$, $\epsilon > 0$ *and* $\delta > 0$, $P_{XZ}(\tilde{\mathcal{E}}_{X|Z}(\hat{F}_{P_{X|Z},n,\lambda_n}) - \tilde{\mathcal{E}}_{X|Z}(F_{P_{X|Z}}) > \epsilon) < \delta$ *for sufficiently large* $n$.

Figure experimentally verifies universal consistency under three noise levels. We use the distributions $Z \sim \mathcal{N}(0, 1)$, (a) $X = e^{-\frac{1}{2}Z^2} \sin(2Z) + N_a$, $N_a \sim 0.3\mathcal{N}(0, 1)$; (b) $X = e^{-\frac{1}{2}Z^2} \sin(2Z) + N_b$, $N_b \sim 3\mathcal{N}(0, 1)$; (c) $X = Z + N_a$, with regularisation $\lambda_n = 10^{-7} n^{-\frac{1}{4}}$.

**Theorem 4.5.** *Assume further that* $F_{P_{X|Z}} \in \mathcal{G}_{\mathcal{X}\mathcal{Z}}$. *Then with probability at least* $1 - \delta$,

$$\tilde{\mathcal{E}}_{X|Z}(\hat{F}_{P_{X|Z},n,\lambda_n}) - \tilde{\mathcal{E}}_{X|Z}(F_{P_{X|Z}}) \leq \lambda_n \left\| F_{P_{X|Z}} \right\|^2_{\mathcal{G}_{\mathcal{X}\mathcal{Z}}}$$

$$+ \frac{2\ln\left(\frac{4}{\delta}\right)}{3n\lambda_n} \left(1 + \sqrt{1 + \frac{18n}{\ln\left(\frac{4}{\delta}\right)}}\right) \left(\left(B_{\mathcal{Z}} \left\| F_{P_{X|Z}} \right\|_{\mathcal{G}_{\mathcal{X}\mathcal{Z}}} + B_{\mathcal{X}}\right)^2 \lambda_n + B_{\mathcal{X}}^2 \left(B_{\mathcal{Z}} + \sqrt{\lambda_n}\right)^2\right)$$

In particular, if $\lambda_n = \mathcal{O}(n^{-1/4})$, then $\tilde{\mathcal{E}}_{X|Z}(\hat{F}_{P_{X|Z},n,\lambda_n}) - \tilde{\mathcal{E}}_{X|Z}(F_{P_{X|Z}}) = \mathcal{O}_p(n^{-1/4})$. The boundedness assumption is satisfied with many commonly used kernels, such as the Gaussian and

Laplacian, and hence is not a restrictive condition. Note that some smoothness assumption on $F_{P_{X|Z}}$ or other distributional assumptions are necessary to achieve universal convergence rates, otherwise the rates can be arbitrarily slow – for more discussion, see e.g. [56, p.56], [11, p.114, Theorem 7.2] or [24, p.32, Theorem 3.1]. It is likely that better (and even optimal) rates can be achieved with further assumptions (see e.g. [5, 53, 3] for results with real or finite-dimensional output spaces), but we leave further investigation of learning rates with infinite-dimensional output spaces as future work.

Theorem 4.4 is stated with respect to the surrogate loss $\tilde{\mathcal{E}}_{X|Z}$, not the original loss $\mathcal{E}_{X|Z}$. Let us now investigate its implications with respect to the original loss. Write $\eta = \tilde{\mathcal{E}}_{X|Z}(F_{P_{X|Z}})$. Since $\tilde{\mathcal{E}}_{X|Z}(\hat{F}_{P_{X|Z},n,\lambda_n}) \geq \mathcal{E}_{X|Z}(\hat{F}_{P_{X|Z},n,\lambda_n})$, a consequence of Theorem 4.4 is that $\lim_{n\to\infty} P_{XZ}(\mathcal{E}_{X|Z}(\hat{F}_{P_{X|Z},n,\lambda_n}) > \epsilon + \eta) \leq \lim_{n\to\infty} P_{XZ}(\tilde{\mathcal{E}}_{X|Z}(\hat{F}_{P_{X|Z},n,\lambda_n}) - \eta > \epsilon) = 0$ for any $\epsilon > 0$. This shows that, in the limit as $n \to \infty$, the loss $\mathcal{E}_{X|Z}(\hat{F}_{P_{X|Z},n,\lambda_n})$ is at most an arbitrarily small amount larger than $\eta$ with high probability.

It remains to investigate what $\eta$ represents, and how large it is. The law of total expectation gives $\eta = \mathbb{E}_{X,Z}[\|k_{\mathcal{X}}(X,\cdot) - F_{P_{X|Z}}(Z)\|_{\mathcal{H}_{\mathcal{X}}}^2] = \mathbb{E}_Z[\mathbb{E}_{X|Z}[\|k_{\mathcal{X}}(X,\cdot) - \mathbb{E}_{X|Z}[k_{\mathcal{X}}(X,\cdot) \mid Z]\|_{\mathcal{H}_{\mathcal{X}}}^2 \mid Z]]$. Here, the integrand $\mathbb{E}_{X|Z}[\|k_{\mathcal{X}}(X,\cdot) - \mathbb{E}_{X|Z}[k_{\mathcal{X}}(X,\cdot)] \mid Z]\|_{\mathcal{H}_{\mathcal{X}}}^2 \mid Z]$ is the *variance* of $k_{\mathcal{X}}(X,\cdot)$ given $Z$ (see [2, p.24] for the definition of the variance of Banach-space valued random variables), and by integrating over $\mathcal{Z}$ in the outer integral, $\eta$ represents the "expected variance" of $k_{\mathcal{X}}(X,\cdot)$.

Suppose $X$ is measurable with respect to $Z$, i.e. $F_{P_{X|Z}}$ has no noise. Then $\mathbb{E}_{X|Z}[k_{\mathcal{X}}(X,\cdot) \mid Z] = k_{\mathcal{X}}(X,\cdot)$, and consequently, $\eta = 0$. In this case, we have universal consistency in both the surrogate loss $\tilde{\mathcal{E}}_{X|Z}$ and the original loss $\mathcal{E}_{X|Z}$. On the other hand, $\eta$ will be large if information about $Z$ tells us little about $X$, and subsequently $k_{\mathcal{X}}(X,\cdot) \in \mathcal{H}_{\mathcal{X}}$. In the extreme case where $X$ and $Z$ are independent, we have $\mathbb{E}_{X|Z}[k_{\mathcal{X}}(X,\cdot) \mid Z] = \mathbb{E}_X[k_{\mathcal{X}}(X,\cdot)]$, and $\eta = \mathbb{E}_X[\|k_{\mathcal{X}}(X,\cdot) - \mathbb{E}_X[k_{\mathcal{X}}(X,\cdot)]\|_{\mathcal{H}_{\mathcal{X}}}^2]$, which is precisely the variance of $k_{\mathcal{X}}(X,\cdot)$ in $\mathcal{H}_{\mathcal{X}}$. Hence, $\eta$ represents the irreducible loss of the true function due to noise in $X$, and the surrogate loss represents the loss functional taking noise into account, while the original loss measures the deviance from the true conditional expectation.

# 5 Measures of discrepancy between conditional distributions and conditional independence

In this section, we propose conditional analogues of the maximum mean discrepancy (MMD) and the Hilbert-Schmidt independence criterion (HSIC), to measure, respectively, the discrepancy between conditional distributions and conditional independence.

## 5.1 Maximum conditional mean discrepancy

Let $X' : \Omega \to \mathcal{X}$, $Z' : \Omega \to \mathcal{Z}$ be additional random variables, with $\int_{\mathcal{X}} \sqrt{k_{\mathcal{X}}(x', x')} dP_{X'}(x') < \infty$. Following Theorem 4.1, we write $\mu_{P_{X|Z}} = F_{P_{X|Z}} \circ Z$ and $\mu_{P_{X'|Z'}} = F_{P_{X'|Z'}} \circ Z'$.

**Definition 5.1.** We define the *maximum conditional mean discrepancy* (MCMD) between $P_{X|Z}$ and $P_{X'|Z'}$ to be the function $\mathcal{Z} \to \mathbb{R}$ defined by $M_{P_{X|Z}, P_{X'|Z'}}(z) = \|F_{P_{X|Z}}(z) - F_{P_{X'|Z'}}(z)\|_{\mathcal{H}_{\mathcal{X}}}$.

Using $\{(x_i, z_i)\}_{i=1}^n, \{(x'_j, z'_j)\}_{j=1}^m$ from joint distributions $P_{XZ}, P_{X'Z'}$, we obtain a closed-form, plug-in estimate from (4) for the square of the MCMD function as

$$\hat{M}_{P_{X|Z}, P_{X'|Z'}}^2(\cdot) = \|\hat{F}_{P_{X|Z},n,\lambda}(\cdot) - \hat{F}_{P_{X'|Z'},m,\lambda'}(\cdot)\|_{\mathcal{H}_{\mathcal{X}}}^2$$
$$= \mathbf{k}_Z^T(\cdot)\mathbf{W}_Z\mathbf{K}_X\mathbf{W}_Z^T\mathbf{k}_Z(\cdot) - 2\mathbf{k}_Z^T(\cdot)\mathbf{W}_Z\mathbf{K}_{XX'}\mathbf{W}_{Z'}^T\mathbf{k}_{Z'}(\cdot) + \mathbf{k}_{Z'}^T(\cdot)\mathbf{W}_{Z'}\mathbf{K}_{X'}\mathbf{W}_{Z'}^T\mathbf{k}_{Z'}(\cdot),$$

where $[\mathbf{K}_X]_{ij} = k_{\mathcal{X}}(x_i, x_j)$, $[\mathbf{K}_{X'}]_{ij} = k_{\mathcal{X}}(x'_i, x'_j)$, $[\mathbf{K}_{XX'}]_{ij} = k_{\mathcal{X}}(x_i, x'_j)$, $[\mathbf{K}_{Z'}]_{ij} = k_{\mathcal{X}}(z'_i, z'_j)$, $\mathbf{k}_{Z'}(\cdot) = (k_Z(z'_1, \cdot), ..., k_Z(z'_m, \cdot))^T$, $\mathbf{W}_Z = (\mathbf{K}_Z + n\lambda\mathbf{I}_n)^{-1}$ and $\mathbf{W}_{Z'} = (\mathbf{K}_{Z'} + m\lambda'\mathbf{I}_m)^{-1}$.

The term MMD stems from the equality $\|\mu_{P_X} - \mu_{P_{X'}}\|_{\mathcal{H}_{\mathcal{X}}} = \sup_{f \in \mathcal{B}_{\mathcal{X}}} |\mathbb{E}_X[f(X)] - \mathbb{E}_{X'}[f(X')]|$ [19, 50], where $\mathcal{B}_{\mathcal{X}} := \{f \in \mathcal{H}_{\mathcal{X}} \mid \|f\|_{\mathcal{H}_{\mathcal{X}}} \leq 1\}$. The supremum is attained by the *witness function*, $\frac{\mu_{P_X} - \mu_{P_{X'}}}{\|\mu_{P_X} - \mu_{P_{X'}}\|_{\mathcal{H}_{\mathcal{X}}}}$ [21]. Using Lemma 3.2, the analogous (almost sure) equality for the MCMD is $\sup_{f \in \mathcal{B}_{\mathcal{X}}} |\mathbb{E}_{X|Z}[f(X) \mid Z] - \mathbb{E}_{X'|Z'}[f(X') \mid Z']| = \|\mu_{P_{X|Z}} - \mu_{P_{X'|Z'}}\|_{\mathcal{H}_{\mathcal{X}}}$. We define

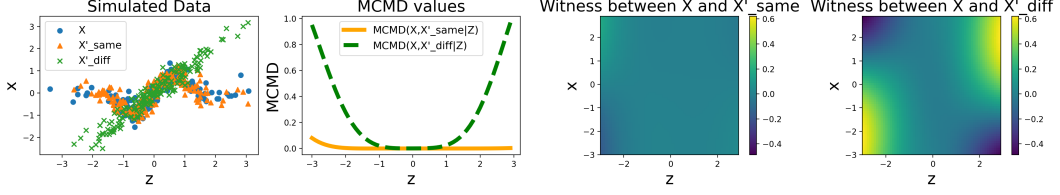

Figure 2: We see that $\mathrm{MCMD}(X, X'_{\mathrm{same}}|Z) \approx 0 \; \forall Z$. Near $Z = 0$, where the dependence on $Z$ of $X$ and $X'_{\mathrm{diff}}$ are similar, $\mathrm{MCMD}(X, X'_{\mathrm{diff}}|Z) \approx 0$, whereas away from 0, the dependence on $Z$ of $X$ and $X'_{\mathrm{diff}}$ are different, and so $\mathrm{MCMD}(X, X'_{\mathrm{diff}}|Z) > 0$. We also see that the conditional witness function between $X$ and $X'_{\mathrm{same}}$ gives 0 at all values of $X$ given any value of $Z$, whereas we have a saddle-like function between $X$ and $X'_{\mathrm{diff}}$, with non-zero functions in $X$ in the regions of $Z$ away from 0.

the *conditional witness function* as the $\mathcal{H}_{\mathcal{X}}$-valued random variable $\frac{\mu_{P_{X|Z}} - \mu_{P_{X'|Z'}}}{\|\mu_{P_{X|Z}} - \mu_{P_{X'|Z'}}\|_{\mathcal{H}_{\mathcal{X}}}}$. We can informally think of $\mathrm{MCMD}_{P_{X|Z}, P_{X'|Z'}}(z)$ as "MMD between $P_{X|Z=z}$ and $P_{X'|Z'=z}$". However, we do not have i.i.d. samples from $P_{X|Z=z}$ and $P_{X'|Z'=z}$, and hence the estimation cannot be done by U- or V-statistic procedures as done for the MMD. The following theorem says that, with characteristic kernels, the MCMD can indeed act as a discrepancy measure between conditional distributions.

**Theorem 5.2.** *Suppose that $k_{\mathcal{X}}$ is characteristic, that $P_Z$ and $P_{Z'}$ are absolutely continuous with respect to each other, and that $P(\cdot \mid Z)$ and $P(\cdot \mid Z')$ admit regular versions. Then $M_{P_{X|Z}, P_{X'|Z'}} = 0$ almost everywhere if and only if, for almost all $z \in \mathcal{Z}$, $P_{X|Z=z}(B) = P_{X'|Z'=z}(B)$ for all $B \in \mathfrak{X}$.*

By [9, p.11 & p.151, Theorem 2.10], we know that the space $(\Omega, \mathcal{F})$ being a Polish space with its Borel $\sigma$-algebra is a sufficient condition for $P(\cdot \mid \mathcal{E})$ to have a regular version for any sub-$\sigma$-algebra $\mathcal{E}$ of $\mathcal{F}$. Hence, the assumption that $P(\cdot \mid Z)$ admits a regular version is not a restrictive one.

The MCMD is reminiscent of the *conditional maximum mean discrepancy* of [39], defined as the Hilbert-Schmidt norm of the operator $\mathcal{U}_{X|Z} - \mathcal{U}_{X'|Z}$ (see Definition 3.4). However, due to previously discussed assumptions, $\mathcal{U}_{X|Z}$ and $\mathcal{U}_{X'|Z}$ often do not even exist, and/or do not have the desired properties of Theorem 3.5, so even at population level, $\mathcal{U}_{X|Z} - \mathcal{U}_{X'|Z}$ is often not an exact measure of discrepancy between conditional distributions, unlike the MCMD. Moreover, [39] only considers the case when the conditioning variable is the same.

## 5.2 Hilbert-Schmidt conditional independence criterion

In this subsection, we introduce a novel criterion of conditional independence.

**Definition 5.3.** We define the *Hilbert-Schmidt Conditional Independence Criterion* between $X$ and $Y$ given $Z$ to be $\mathrm{HSCIC}(X, Y \mid Z) = \|\mu_{P_{XY|Z}} - \mu_{P_{X|Z}} \otimes \mu_{P_{Y|Z}}\|_{\mathcal{H}_{\mathcal{X}} \otimes \mathcal{H}_{\mathcal{Y}}}$.

We can write $\mathrm{HSCIC}(X, Y \mid Z) = H_{X,Y|Z} \circ Z$ for some $H_{X,Y|Z} : \mathcal{Z} \to \mathbb{R}$. Given a sample $\{(x_i, y_i, z_i)\}_{i=1}^n$ from $P_{XYZ}$, we obtain a plug-in, closed-form estimate of $H^2_{X,Y|Z}(\cdot)$ as follows:

$$\hat{H}^2_{X,Y|Z}(\cdot) = \mathbf{k}_Z^T(\cdot)\mathbf{W}(\mathbf{K}_X \odot \mathbf{K}_Y)\mathbf{W}^T\mathbf{k}_Z(\cdot) - 2\mathbf{k}_Z^T(\cdot)\mathbf{W}((\mathbf{K}_X\mathbf{W}^T\mathbf{k}_Z(\cdot)) \odot (\mathbf{K}_Y\mathbf{W}^T\mathbf{k}_Z(\cdot)))$$
$$+ (\mathbf{k}_Z^T(\cdot)\mathbf{W}\mathbf{K}_X\mathbf{W}^T\mathbf{k}_Z(\cdot))(\mathbf{k}_Z^T(\cdot)\mathbf{W}\mathbf{K}_Y\mathbf{W}^T\mathbf{k}_Z(\cdot))$$

where $[\mathbf{K}_Y]_{ij} := k_{\mathcal{Y}}(y_i, y_j)$ and $\odot$ denotes elementwise multiplication of matrices.

Casting aside measure-theoretic issues arising from conditioning on an event of probability 0, we can conceptually think of the realisation of the HSCIC at each $z = Z(\omega)$ as "the HSIC between $P_{X|Z=z}$ and $P_{Y|Z=z}$". Again, we do not have multiple samples from each distribution $P_{X|Z=z}$ and $P_{Y|Z=z}$, so the estimation cannot be done by U- or V-statistic procedures as done for HSIC. The following theorem shows that HSCIC is a measure of conditional independence.

**Theorem 5.4.** *Suppose $k_{\mathcal{X}} \otimes k_{\mathcal{Y}}$ is a characteristic kernel[3] on $\mathcal{X} \times \mathcal{Y}$, and that $P(\cdot \mid Z)$ admits a regular version. Then $\mathrm{HSCIC}(X, Y \mid Z) = 0$ almost surely if and only if $X \perp\!\!\!\perp Y \mid Z$.*

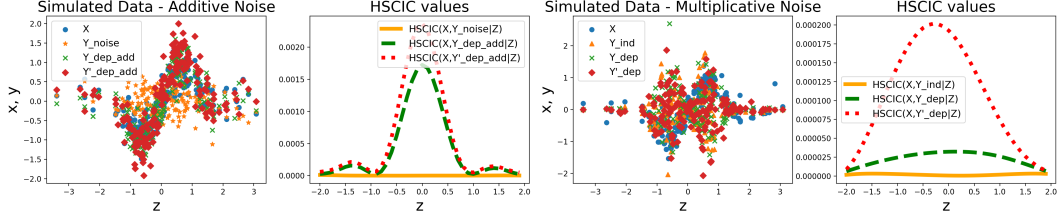

Figure 3: We see that $\text{HSCIC}(X, Y_{\text{noise}}|Z) \approx 0$ (left) and $\text{HSCIC}(X, Y_{\text{ind}}|Z) \approx 0$ (right) for all $Z$, whereas $\text{HSCIC}(X, Y_{\text{dep\_add}}|Z) > 0$, $\text{HSCIC}(X, Y'_{\text{dep\_add}}|Z) > 0$, $\text{HSCIC}(X, Y_{\text{dep}}|Z) > 0$, $\text{HSCIC}(X, Y'_{\text{dep}}|Z) > 0$. In particular, the dependence of $Y'_{\text{dep\_add}}$ and $Y'_{\text{dep}}$ on $X$ is greater than that of $Y_{\text{dep\_add}}$ and $Y_{\text{dep}}$, and is represented by larger values of $\text{HSCIC}(X, Y'_{\text{dep\_add}}|Z)$ and $\text{HSCIC}(X, Y'_{\text{dep}}|Z)$ compared to $\text{HSCIC}(X, Y_{\text{dep}}|Z)$ and $\text{HSCIC}(X, Y_{\text{dep\_add}}|Z)$.

Concurrent and independent work by Sheng and Sriperumbudur [43] proposes a similar criterion with the same nomenclature (HSCIC). However, they omit the discussion of CMEs entirely, and define the HSCIC as the usual HSIC between $P_{XY|Z=z}$ and $P_{X|Z=z}P_{Y|Z=z}$, without considerations for conditioning on an event of measure 0. Their focus is more on investigating connections to distance-based measures [57, 42]. Fukumizu et al. [16] propose $I^{COND}$, defined as the squared Hilbert-Schmidt norm of the normalised conditional cross-covariance operator $V_{\ddot{Y}\ddot{X}|Z} := \mathcal{C}_{\ddot{Y}\ddot{Y}}^{-1/2} \Sigma_{\ddot{Y}\ddot{X}|Z} \mathcal{C}_{\ddot{X}\ddot{X}}^{-1/2}$, where $\ddot{X} := (X, Z)$ and $\ddot{Y} := (Y, Z)$. As discussed, these operator-based definitions rely on a number of strong assumptions that will often mean that $V_{\ddot{Y}\ddot{X}|Z}$ does not exist, or it does not satisfy the conditions for it to be used as an exact criterion even at population level. On the other hand, the HSCIC defined as in Definition 5.3 is an exact mathematical criterion of conditional independence at population level. Note that $I^{COND}$ is a single-value criterion, whereas the HSCIC is a random criterion.

## 5.3 Experiments

We carry out simulations to demonstrate the behaviour of the MCMD and HSCIC. In all simulations, we use the Gaussian kernel $k_{\mathcal{X}}(x, x') = k_{\mathcal{Y}}(x, x') = k_{\mathcal{Z}}(x, x') = e^{-\frac{1}{2}\sigma_X \|x - x'\|_2^2}$ with hyperparameter $\sigma_X = 0.1$, and regularisation parameter $\lambda = 0.01$.

In Figure 2, we simulate 500 samples from $Z, Z' \sim \mathcal{N}(0, 1)$, $X = e^{-0.5Z^2} \sin(2Z) + N_X$, $X'_{\text{same}} = e^{-0.5Z'^2} \sin(2Z') + N_X$ and $X'_{\text{diff}} = Z' + N_X$, where $N_X \sim 0.3\mathcal{N}(0, 1)$ is the (additive) noise variable. The first plot shows simulated data, the second MCMD values against Z, and the heatmaps show the (unnormalised) conditional witness function, whose norm gives the MCMD.

In Figure 3, on the left, we simulate 500 samples from the additive noise model, $Z \sim \mathcal{N}(0, 1)$, $X = e^{-0.5Z^2} \sin(2Z) + N_X$, $Y_{\text{noise}} = N_Y$, $Y_{\text{dep\_add}} = e^{-0.5Z^2} \sin(2Z) + N_X + 0.2X$ and $Y'_{\text{dep\_add}} = e^{-0.5Z^2} \sin(2Z) + N_X + 0.4X$, where $N_X \sim 0.3\mathcal{N}(0, 1)$ is the (additive) noise variable. On the right, we simulate 500 samples from the multiplicative noise model, $Z \sim \mathcal{N}(0, 1)$, $X = Y_{\text{ind}} = e^{-0.5Z^2} \sin(2Z)N_X$, $Y_{\text{dep}} = e^{-0.5Z^2} \sin(2Z)N_Y + 0.2X$ and $Y'_{\text{dep}} = e^{-0.5Z^2} \sin(2Z)N_Y + 0.4X$, where $N_X, N_Y \sim 0.3\mathcal{N}(0, 1)$ are the (multiplicative) noise variables.

## 6 Conclusion

In this paper, we proposed a new approach to kernel conditional mean embeddings, based on Bochner conditional expectation. Compared to the previous operator-based approaches, it does not rely on stringent assumptions that are often violated in common situations. Using this new approach, we discussed how to obtain empirical estimates via natural vector-valued regression, establishing universal consistency under no distributional assumptions and convergence rate of $\mathcal{O}_p(n^{-1/4})$ in the well-specified case. Finally, we extended the notions of the MMD, witness function and HSIC to the conditional case. We believe that our new approach has the potential to unlock the powerful arsenal of kernel mean embeddings to the conditional setting, in a more convenient and rigorous manner.

## Broader Impact

The nature of this work is theoretical, and hence we do not feel it is applicable to discuss its broader societal impact.

## Acknowledgments and Disclosure of Funding

We thank Mattes Mollenhauer at Freie Universität Berlin for pointing out the missing conditions on the regularization parameter of our initial universal consistency result, and for other fruitful discussions. We also thank anonymous reviewers for pointing out typos, suggesting several improvements and correcting a mistake in the proof of Theorem 4.1. Finally, we thank Simon Buchholz, Alessandro Ialongo, Heiner Kremer and Jonas Kübler at MPI Tübingen for helpful feedback on initial drafts.

The idea behind this paper was conceived, and part of the work done, while JP was a Master's student at the Seminar for Statistics, Department of Mathematics, ETH Zürich. JP is extremely grateful to his Master's thesis supervisor, Professor Sara van de Geer, for readily accepting the proposed topic, and her expert guidance throughout the thesis.

This work was funded by the federal and state governments of Germany through the Max Planck Society (MPG).

## Footnotes

[1]Here, the term "kernel" must not be confused with the kernel associated to RKHSs.

[2]$\mathcal{E}_{X|Z}$ is not the only loss function, nor is $l_{\mathcal{X}\mathcal{Z}}$ the only kernel, that we can use for this problem. Kadri et al. [26] discuss various operator-valued kernels that can be used (albeit without closed-form solutions) and Laforgue et al. [28] discuss other loss functions that can be used for more robust estimates. We view this flexibility to facilitate other loss and kernel functions in the regression set-up, although not explored in depth in this work, as a significant advantage over the previous approaches.

[3]See [54] for a detailed discussion on characteristic tensor product kernels.

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
