[Supplementary Material]

# A Generalised Jensen's Inequality

In Section 4, we require a version of Jensen's inequality generalised to (possibly) infinite-dimensional vector spaces, because our random variable takes values in $\mathcal{H}_\mathcal{X}$, and our convex function is $\|\cdot\|_{\mathcal{H}_\mathcal{X}}^2 : \mathcal{H}_\mathcal{X} \to \mathbb{R}$. Note that this square norm function is indeed convex, since, for any $t \in [0,1]$ and any pair $f, g \in \mathcal{H}_\mathcal{X}$,

$$\|tf + (1-t)g\|_{\mathcal{H}_\mathcal{X}}^2 \leq (t\|f\|_{\mathcal{H}_\mathcal{X}} + (1-t)\|g\|_{\mathcal{H}_\mathcal{X}})^2 \qquad \text{by the triangle inequality}$$
$$\leq t\|f\|_{\mathcal{H}_\mathcal{X}}^2 + (1-t)\|g\|_{\mathcal{H}_\mathcal{X}}^2, \qquad \text{by the convexity of } x \mapsto x^2.$$

The following theorem generalises Jensen's inequality to infinite-dimensional vector spaces.

**Theorem A.1** (Generalised Jensen's Inequality, [38], Theorem 3.10). *Suppose $\mathcal{T}$ is a real Hausdorff locally convex (possibly infinite-dimensional) linear topological space, and let $C$ be a closed convex subset of $\mathcal{T}$. Suppose $(\Omega, \mathcal{F}, P)$ is a probability space, and $V : \Omega \to \mathfrak{T}$ a Pettis-integrable random variable such that $V(\Omega) \subseteq C$. Let $f : C \to [-\infty, \infty)$ be a convex, lower semi-continuous extended-real-valued function such that $\mathbb{E}_V[f(V)]$ exists. Then*

$$f(\mathbb{E}_V[V]) \leq \mathbb{E}_V[f(V)].$$

We will actually apply generalised Jensen's inequality with conditional expectations, so we need the following theorem.

**Theorem A.2** (Generalised Conditional Jensen's Inequality). *Suppose $\mathcal{T}$ is a real Hausdorff locally convex (possibly infinite-dimensional) linear topological space, and let $C$ be a closed convex subset of $\mathcal{T}$. Suppose $(\Omega, \mathcal{F}, P)$ is a probability space, and $V : \Omega \to \mathcal{T}$ a Pettis-integrable random variable such that $V(\Omega) \subseteq C$. Let $f : C \to [-\infty, \infty)$ be a convex, lower semi-continuous extended-real-valued function such that $\mathbb{E}_V[f(V)]$ exists. Suppose $\mathcal{E}$ is a sub-$\sigma$-algebra of $\mathcal{F}$. Then*

$$f(\mathbb{E}[V \mid \mathcal{E}]) \leq \mathbb{E}[f(V) \mid \mathcal{E}].$$

*Proof.* Let $\mathcal{T}^*$ be the dual space of all real-valued continuous linear functionals on $\mathcal{T}$. The first part of the proof of [38, Theorem 3.6] tells us that, for all $v \in \mathcal{T}$, we can write

$$f(v) = \sup\{m(v) \mid m \text{ affine}, m \leq f \text{ on } C\},$$

where an *affine* function $m$ on $\mathcal{T}$ is of the form $m(v) = v^*(v) + \alpha$ for some $v^* \in \mathcal{T}^*$ and $\alpha \in \mathbb{R}$. If we define the subset $Q$ of $\mathcal{T}^* \times \mathbb{R}$ as

$$Q := \{(v^*, \alpha) : v^* \in \mathcal{T}^*, \alpha \in \mathbb{R}, v^*(v) + \alpha \leq f(v) \text{ for all } v \in \mathcal{T}\},$$

then we can rewrite $f$ as

$$f(v) = \sup_{(v^*, \alpha) \in Q} \{v^*(v) + \alpha\}, \qquad \text{for all } v \in \mathcal{T}. \tag{5}$$

See that, for any $(v^*, \alpha) \in Q$, we have

$$\mathbb{E}\left[f(V) \mid \mathcal{E}\right] \geq \mathbb{E}\left[v^*(V) + \alpha \mid \mathcal{E}\right] \qquad \text{almost surely, by assumption (*)}$$
$$= \mathbb{E}\left[v^*(V) \mid \mathcal{E}\right] + \alpha \qquad \text{almost surely, by linearity (**).}$$

Here, (*) and (**) use the properties of conditional expectation of vector-valued random variables given in [12, pp.45-46, Properties 43 and 40 respectively].

We want to show that $\mathbb{E}\left[v^*(V) \mid \mathcal{E}\right] = v^*\left(\mathbb{E}\left[V \mid \mathcal{E}\right]\right)$ almost surely, and in order to so, we show that the right-hand side is a version of the left-hand side. The right-hand side is clearly $\mathcal{E}$-measurable, since we have a linear operator on an $\mathcal{E}$-measurable random variable. Moreover, for any $A \in \mathcal{E}$,

$$\int_A v^*\left(\mathbb{E}\left[V \mid \mathcal{E}\right]\right) dP = v^*\left(\int_A \mathbb{E}\left[V \mid \mathcal{E}\right] dP\right) \qquad \text{by [10, p.403, Proposition E.11]}$$

$$= v^*\left(\int_A V dP\right) \qquad \text{by the definition of conditional expectation}$$

$$= \int_A v^*(V) dP \qquad \text{by [10, p.403, Proposition E.11]}$$

(here, all the equalities are almost-sure equalities). Hence, by the definition of the conditional expectation, we have that $\mathbb{E}\left[v^*(V) \mid \mathcal{E}\right] = v^*\left(\mathbb{E}\left[V \mid \mathcal{E}\right]\right)$ almost surely. Going back to our above work, this means that

$$\mathbb{E}\left[f(V) \mid \mathcal{E}\right] \geq v^*\left(\mathbb{E}\left[V \mid \mathcal{E}\right]\right) + \alpha.$$

Now take the supremum of the right-hand side over $Q$. Then (5) tells us that

$$\mathbb{E}\left[f(V) \mid \mathcal{E}\right] \geq f\left(\mathbb{E}\left[V \mid \mathcal{E}\right]\right),$$

as required. $\qquad\square$

In the context of Section 4, $\mathcal{H}_{\mathcal{X}}$ is real and Hausdorff, and locally convex (because it is a normed space). We take the closed convex subset to be the whole space $\mathcal{H}_{\mathcal{X}}$ itself. The function $\|\cdot\|_{\mathcal{H}_{\mathcal{X}}}^2 : \mathcal{H}_{\mathcal{X}} \to \mathbb{R}$ is convex (as shown above) and continuous, and finally, since Bochner-integrability implies Pettis integrability, all the conditions of Theorem A.2 are satisfied.

## B  Generalisation Error Bounds

Caponnetto and De Vito [5] give an optimal rate of convergence of vector-valued RKHS regression estimators, and its results are quoted by Grünewälder et al. [22] as the state of the art convergence rates, $O(\frac{\log n}{n})$. In particular, this implies that the learning algorithm is consistent. However, the lower rate uses an assumption that the output space is a finite-dimensional Hilbert space [5, Theorem 2]; and in our case, this will mean that $\mathcal{H}_{\mathcal{X}}$ is finite-dimensional. This is not true if, for example, we take $k_{\mathcal{X}}$ to be the Gaussian kernel; indeed, this is noted as a limitation by Grünewälder et al. [22], stating that "It is likely that this (finite-dimension) assumption can be weakened, but this requires a deeper analysis". In this paper, we do not want to restrict our attention to finite-dimensional $\mathcal{H}_{\mathcal{X}}$. The upper bound would have been sufficient to guarantee consistency, but an assumption used in the upper bound requires the operator $l_{XZ,z} : \mathcal{H}_{\mathcal{X}} \to \mathcal{G}_{\mathcal{X}\mathcal{Z}}$ defined by

$$l_{XZ,z}(f)(z') = l_{XZ}(z, z')(f)$$

to be Hilbert-Schmidt for all $z \in \mathcal{Z}$. However, for each $z \in \mathcal{Z}$, taking any orthonormal basis $\{\varphi_i\}_{i=1}^{\infty}$ of $\mathcal{H}_{\mathcal{X}}$, we see that

$$\begin{aligned}
\sum_{i=1}^{\infty} \langle l_{XZ,z}(\varphi_i), l_{XZ,z}(\varphi_i) \rangle_{\mathcal{G}_{\mathcal{X}\mathcal{Z}}} &= \sum_{i=1}^{\infty} \langle k_{\mathcal{Z}}(z, \cdot)\varphi_i, k_{\mathcal{Z}}(z, \cdot)\varphi_i \rangle_{\mathcal{G}_{\mathcal{X}\mathcal{Z}}} \\
&= \sum_{i=1}^{\infty} \langle k_{\mathcal{Z}}(z, z)\varphi_i, \varphi_i \rangle_{\mathcal{H}_{\mathcal{X}}} \\
&= k_{\mathcal{Z}}(z, z) \sum_{i=1}^{\infty} 1 \\
&= \infty,
\end{aligned}$$

meaning this assumption is not fulfilled with our choice of kernel either. Hence, results in [5], used by [22], are not applicable to guarantee consistency in our context.

Kadri et al. [26] address the problem of generalisability of function-valued learning algorithms, using the concept of uniform algorithmic stability [4]. Let us write

$$\mathcal{D} := \{(x_1, z_1), ..., (x_n, z_n)\}$$

for our training set of size $n$ drawn i.i.d. from the distribution $P_{XZ}$, and we denote by $\mathcal{D}^i = \mathcal{D}\backslash(x_i, z_i)$ the set $\mathcal{D}$ from which the data point $(x_i, z_i)$ is removed. Further, we denote by $\hat{F}_{P_{X|Z},\mathcal{D}} = \hat{F}_{P_{X|Z},n,\lambda}$ the estimate produced by our learning algorithm from the dataset $\mathcal{D}$ by minimising the loss $\hat{\mathcal{E}}_{X|Z,n,\lambda}(F) = \sum_{i=1}^{n} \|k_{\mathcal{X}}(x_i, \cdot) - F(z_i)\|_{\mathcal{H}_{\mathcal{X}}}^2 + \lambda\|F\|_{\mathcal{G}_{\mathcal{X}\mathcal{Z}}}^2$

The assumptions used in this paper, with notations translated to our context, are

1. There exists $\kappa_1 > 0$ such that for all $z \in \mathcal{Z}$,

$$\|l_{\mathcal{X}\mathcal{Z}}(z,z)\|_{\text{op}} = \sup_{f \in \mathcal{H}_{\mathcal{X}}} \frac{\|l_{\mathcal{X}\mathcal{Z}}(z,z)(f)\|_{\mathcal{H}_{\mathcal{X}}}}{\|f\|_{\mathcal{H}_{\mathcal{X}}}} \leq \kappa_1^2.$$

2. The real function $\mathcal{Z} \times \mathcal{Z} \to \mathbb{R}$ defined by

$$(z_1, z_2) \mapsto \langle l_{\mathcal{X}\mathcal{Z}}(z_1, z_2) f_1, f_2 \rangle_{\mathcal{H}_{\mathcal{X}}}$$

is measurable for all $f_1, f_2 \in \mathcal{H}_{\mathcal{X}}$.

3. The map $(f, F, z) \mapsto \|f - F(z)\|_{\mathcal{H}_{\mathcal{X}}}^2$ is $\tau$-admissible, i.e. convex with respect to $F$ and Lipschitz continuous with respect to $F(z)$, with $\tau$ as its Lipschitz constant.

4. There exists $\kappa_2 > 0$ such that for all $(z, f) \in \mathcal{Z} \times \mathcal{H}_{\mathcal{X}}$ and any training set $\mathcal{D}$,

$$\|f - \hat{F}_{P_{X|Z},\mathcal{D}}(z)\|_{\mathcal{H}_{\mathcal{X}}}^2 \leq \kappa_2.$$

The concept of *uniform stability*, with notations translated to our context, is defined as follows.

**Definition B.1** (Uniform algorithmic stability, [26, Definition 6]). For each $F \in \mathcal{G}_{\mathcal{X}\mathcal{Z}}$, define the function

$$\mathcal{R}(F) : \mathcal{Z} \times \mathcal{H}_{\mathcal{X}} \to \mathbb{R}$$
$$(z, x) \mapsto \|k_{\mathcal{X}}(x, \cdot) - F(z)\|_{\mathcal{H}_{\mathcal{X}}}^2.$$

A learning algorithm that calculates the estimate $\hat{F}_{P_{X|Z},\mathcal{D}}$ from a training set has uniform stability $\beta$ with respect to the squared loss if the following holds: for all $n \geq 1$, all $i \in \{1, ..., n\}$ and any training set $\mathcal{D}$ of size $n$,

$$\|\mathcal{R}(\hat{F}_{P_{X|Z},\mathcal{D}}) - \mathcal{R}(\hat{F}_{P_{X|Z},\mathcal{D}^i})\|_\infty \leq \beta.$$

The next two theorems are quoted from [26].

**Theorem B.2** ([26, Theorem 7]). *Under assumptions 1, 2 and 3, a learning algorithm that maps a training set $\mathcal{D}$ to the function $\hat{F}_{P_{X|Z},\mathcal{D}} = \hat{F}_{P_{X|Z},n,\lambda}$ is $\beta$-stable with*

$$\beta = \frac{\tau^2 \kappa_1^2}{2\lambda n}.$$

**Theorem B.3** ([26, Theorem 8]). *Let $\mathcal{D} \mapsto \hat{F}_{P_{X|Z},\mathcal{D}} = \hat{F}_{P_{X|Z},n,\lambda}$ be a learning algorithm with uniform stability $\beta$, and assume Assumption 4 is satisfied. Then, for all $n \geq 1$ and any $0 < \delta < 1$, the following bound holds with probability at least $1 - \delta$ over the random draw of training samples:*

$$\tilde{\mathcal{E}}_{X|Z}(\hat{F}_{P_{X|Z},n,\lambda}) \leq \frac{1}{n} \hat{\mathcal{E}}_{X|Z,n}(\hat{F}_{P_{X|Z},n,\lambda}) + 2\beta + (4n\beta + \kappa_2) \sqrt{\frac{\ln \frac{1}{\delta}}{2n}}.$$

Theorems B.2 and B.3 give us results about the generalisability of our learning algorithm. It remains to check whether the assumptions are satisfied.

Assumption 2 is satisfied thanks to our assumption that point embeddings are measurable functions, and Assumption 1 is satisfied if we assume that $k_{\mathcal{Z}}$ is a bounded kernel (i.e. there exists $B_{\mathcal{Z}} > 0$ such that $k_{\mathcal{Z}}(z_1, z_2) \leq B_{\mathcal{Z}}$ for all $z_1, z_2 \in \mathcal{Z}$), because

$$\|l_{\mathcal{X}\mathcal{Z}}(z,z)\|_{\text{op}} = \sup_{f \in \mathcal{H}_{\mathcal{X}}, \|f\|_{\mathcal{H}_{\mathcal{X}}}=1} \|k_{\mathcal{Z}}(z,z)(f)\|_{\mathcal{H}_{\mathcal{X}}} \leq B_{\mathcal{Z}}.$$

In [26], a general loss function is used rather than the squared loss, and it is noted that Assumption 3 is in general *not* satisfied with the squared loss, which is what we use in our context. However, this issue can be addressed if we restrict the output space to a bounded subset. In fact, the only elements in $\mathcal{H}_{\mathcal{X}}$ that appear as the output samples in our case are $k_{\mathcal{X}}(x, \cdot)$ for $x \in \mathcal{X}$, so if we place the assumption that $k_{\mathcal{X}}$ is a bounded kernel (i.e. there exists $B_{\mathcal{X}} > 0$ such that $k_{\mathcal{X}}(x_1, x_2) \leq B_{\mathcal{X}}$ for all $x_1, x_2 \in \mathcal{X}$), then by the reproducing property,

$$\|k_{\mathcal{X}}(x, \cdot)\|_{\mathcal{H}_{\mathcal{X}}} = \sqrt{k_{\mathcal{X}}(x, x)} \leq \sqrt{B_{\mathcal{X}}}.$$

So it is no problem, in our case, to place this boundedness assumption. [26, Appendix D] tells us that Assumption 1 with this boundedness assumption implies Assumption 4 with

$$\kappa_2 = B_{\mathcal{X}} \left( 1 + \frac{\kappa_1}{\sqrt{\lambda}} \right)^2,$$

while [26, Lemma 2] provides us with a condition which can replace Assumption 3 in Theorem B.2, giving us the uniform stability of our algorithm with

$$\beta = \frac{2\kappa_1^2 B_{\mathcal{X}} \left( 1 + \frac{\kappa_1}{\sqrt{\lambda}} \right)^2}{\lambda n}.$$

Then the result of Theorem B.3 holds with this new $\beta$.

## C  Proofs

**Lemma 2.1.** For each $f \in \mathcal{H}_{\mathcal{X}}$, $\int_{\mathcal{X}} f(x) dP_X(x) = \langle f, \mu_{P_X} \rangle_{\mathcal{H}_{\mathcal{X}}}$.

*Proof.* Let $L_P$ be a functional on $\mathcal{H}$ defined by $L_P(f) := \int_{\mathcal{X}} f(x) dP(x)$. Then $L_P$ is clearly linear, and moreover,

$$
\begin{aligned}
|L_P(f)| &= \left| \int_{\mathcal{X}} f(x) dP(x) \right| \\
&= \left| \int_{\mathcal{X}} \langle f, k(x, \cdot) \rangle_{\mathcal{H}} dP(x) \right| && \text{by the reproducing property} \\
&\leq \int_{\mathcal{X}} |\langle f, k(x, \cdot) \rangle_{\mathcal{H}}| dP(x) && \text{by Jensen's inequality} \\
&\leq \|f\|_{\mathcal{H}} \int_{\mathcal{X}} \|k(x, \cdot)\|_{\mathcal{H}} dP(x) && \text{by Cauchy-Schwarz inequalty.}
\end{aligned}
$$

Since the map $x \mapsto k(x, \cdot)$ is Bochner $P$-integrable, $L_P$ is bounded, i.e. $L_P \in \mathcal{H}^*$. So by the Riesz Representation Theorem, there exists a unique $h \in \mathcal{H}$ such that $L_P(f) = \langle f, h \rangle_{\mathcal{H}}$ for all $f \in \mathcal{H}$.

Choose $f(\cdot) = k(x, \cdot)$ for some $x \in \mathcal{X}$. Then

$$
\begin{aligned}
h(x) &= \langle k(x, \cdot), h \rangle_{\mathcal{H}} \\
&= L_P(k(x, \cdot)) \\
&= \int_{\mathcal{X}} k(x', x) dP(x'),
\end{aligned}
$$

which means $h(\cdot) = \int_{\mathcal{X}} k(x, \cdot) dP(x) = \mu_P(\cdot)$ (implicitly applying [12, Corollary 37]). □

**Lemma 2.3.** For $f \in \mathcal{H}_{\mathcal{X}}, g \in \mathcal{H}_{\mathcal{Y}}$, $\langle f \otimes g, \mu_{P_{XY}} \rangle_{\mathcal{H}_{\mathcal{X}} \otimes \mathcal{H}_{\mathcal{Y}}} = \mathbb{E}_{XY}[f(X)g(Y)]$.

*Proof.* For Bochner integrability, we see that

$$
\begin{aligned}
\mathbb{E}_{XY} \left[ \left\| k_{\mathcal{X}}(X, \cdot) \otimes k_{\mathcal{Y}}(Y, \cdot) \right\|_{\mathcal{H}_{\mathcal{X}} \otimes \mathcal{H}_{\mathcal{Y}}} \right] &= \mathbb{E}_{XY} \left[ \sqrt{k_{\mathcal{X}}(X, X)} \sqrt{k_{\mathcal{Y}}(Y, Y)} \right] \\
&\leq \sqrt{\mathbb{E}_X \left[ k_{\mathcal{X}}(X, X) \right]} \sqrt{\mathbb{E}_Y \left[ k_{\mathcal{Y}}(Y, Y) \right]},
\end{aligned}
$$

by Cauchy-Schwarz inequality. (2) now implies that $k_{\mathcal{X}}(X, \cdot) \otimes k_{\mathcal{Y}}(Y, \cdot)$ is Bochner $P_{XY}$-integrable.

Let $L_{P_{XY}}$ be a functional on $\mathcal{H}_{\mathcal{X}} \otimes \mathcal{H}_{\mathcal{Y}}$ defined by $L_{P_{XY}} \left( \sum_i f_i \otimes g_i \right) := \mathbb{E}_{XY} \left[ \sum_i f_i(X) g_i(Y) \right]$. Then $L_{P_{XY}}$ is clearly linear, and moreover,

$$
\begin{aligned}
|L_{P_{XY}}(\sum_i f_i \otimes g_i)| &= |\mathbb{E}_{XY}[\sum_i f_i(X) g_i(Y)]| \\
&\leq \mathbb{E}_{XY}[|\sum_i f_i(X) g_i(Y)|] && \text{by Jensen's inequality}
\end{aligned}
$$

$$= \mathbb{E}_{XY}[|\langle \sum_i f_i \otimes g_i, k_{\mathcal{X}}(X, \cdot) \otimes k_{\mathcal{Y}}(Y, \cdot)\rangle_{\mathcal{H}_{\mathcal{X}} \otimes \mathcal{H}_{\mathcal{Y}}}|] \qquad \text{by the reproducing property}$$

$$\leq \|\sum_i f_i \otimes g_i\|_{\mathcal{H}_{\mathcal{X}} \otimes \mathcal{H}_{\mathcal{Y}}} \mathbb{E}_{XY}\left[\|k_{\mathcal{X}}(X, \cdot) \otimes k_{\mathcal{Y}}(Y, \cdot)\|_{\mathcal{H}_{\mathcal{X}} \otimes \mathcal{H}_{\mathcal{Y}}}\right] \qquad \text{by Cauchy-Schwarz inequality.}$$

Hence, by Bochner integrability shown above, $L_{P_{XY}} \in (\mathcal{H}_{\mathcal{X}} \otimes \mathcal{H}_{\mathcal{Y}})^*$. So by the Riesz Representation Theorem, there exists $h \in \mathcal{H}_{\mathcal{X}} \otimes \mathcal{H}_{\mathcal{Y}}$ such that $L_{P_{XY}}(\sum_i f_i \otimes g_i) = \langle \sum_i f_i \otimes g_i, h\rangle_{\mathcal{H}_{\mathcal{X}} \otimes \mathcal{H}_{\mathcal{Y}}}$ for all $\sum_i f_i \otimes g_i \in \mathcal{H}_{\mathcal{X}} \otimes \mathcal{H}_{\mathcal{Y}}$.

Choose $k_{\mathcal{X}}(x, \cdot) \otimes k_{\mathcal{Y}}(y, \cdot) \in \mathcal{H}_{\mathcal{X}} \otimes \mathcal{H}_{\mathcal{Y}}$ for some $x \in \mathcal{X}$ and $y \in \mathcal{Y}$. Then

$$\begin{aligned} h(x, y) &= \langle k_{\mathcal{X}}(x, \cdot) \otimes k_{\mathcal{Y}}(y, \cdot), h\rangle_{\mathcal{H}_{\mathcal{X}} \otimes \mathcal{H}_{\mathcal{Y}}} \qquad \text{by the reproducing property} \\ &= L_{P_{XY}}(k_{\mathcal{X}}(x, \cdot) \otimes k_{\mathcal{Y}}(y, \cdot)) \\ &= \mathbb{E}_{XY}\left[k_{\mathcal{X}}(x, X) \otimes k_{\mathcal{Y}}(y, Y)\right] \\ &= \mu_{P_{XY}}(x, y), \end{aligned}$$

as required. $\qquad \square$

**Lemma C.1.** *Let $\{\varphi_i\}_{i=1}^{\infty}$ and $\{\psi_j\}_{j=1}^{\infty}$ be orthonormal bases of $\mathcal{H}_{\mathcal{X}}$ and $\mathcal{H}_{\mathcal{Y}}$ respectively (note that they are countable, since the RKHSs are separable). Then the map*

$$\Phi : \mathcal{H}_{\mathcal{X}} \otimes \mathcal{H}_{\mathcal{Y}} \to HS(\mathcal{H}_{\mathcal{X}}, \mathcal{H}_{\mathcal{Y}})$$

$$\sum_{i=1, j=1}^{\infty} c_{i,j}(\varphi_i \otimes \psi_j) \mapsto [h \mapsto \sum_{i=1, j=1}^{\infty} c_{i,j}\langle h, \varphi_i\rangle_{\mathcal{H}_{\mathcal{X}}} \psi_j]$$

*is an isometric isomorphism.*

*Proof.* $\Phi$ is clearly linear. We first show isometry:

$$\begin{aligned} \left\|\Phi\left(\sum_{i=1, j=1}^{\infty} c_{i,j}(\varphi_i \otimes \psi_j)\right)\right\|_{\text{HS}}^2 &= \left\|\sum_{i=1, j=1}^{\infty} c_{i,j}\langle \cdot, \varphi_i\rangle_{\mathcal{H}_{\mathcal{X}}} \psi_j\right\|_{\text{HS}}^2 \\ &= \sum_{k=1}^{\infty} \left\|\sum_{i=1, j=1}^{\infty} c_{i,j}\langle \varphi_k, \varphi_i\rangle_{\mathcal{H}_{\mathcal{X}}} \psi_j\right\|_{\mathcal{H}_{\mathcal{Y}}}^2 \qquad \text{by definition} \\ &= \sum_{i=1, j=1}^{\infty} c_{i,j}^2 \qquad \text{by orthonormality} \\ &= \left\|\sum_{i=1, j=1}^{\infty} c_{i,j}(\varphi_i \otimes \psi_j)\right\|_{\mathcal{H}_{\mathcal{X}} \otimes \mathcal{H}_{\mathcal{Y}}}^2 \qquad \text{by orthonormality,} \end{aligned}$$

as required. It remains to show surjectivity.

Take an element $T \in \text{HS}(\mathcal{H}_{\mathcal{X}}, \mathcal{H}_{\mathcal{Y}})$. Then $T$ is completely determined by $\{T\varphi_i\}_{i=1}^{\infty}$. For each $i$, suppose $T\varphi_i = \sum_{j=1}^{\infty} d_j^i \psi_j$, with $d_j^i \in \mathbb{R}$ for all $i$ and $j$. Then

$$\begin{aligned} \Phi\left(\sum_{i'=1, j=1}^{\infty} d_j^{i'}(\varphi_{i'} \otimes \psi_j)\right) &= \left[\varphi_i \mapsto \sum_{i'=1, j=1}^{\infty} \langle d_j^{i'}\varphi_{i'}, \varphi_i\rangle_{\mathcal{H}_{\mathcal{X}}} \psi_j\right] \\ &= \left[\varphi_i \mapsto \sum_{j=1}^{\infty} d_j^i \psi_j\right] \qquad \text{by orthonormality} \\ &= T. \end{aligned}$$

So $\Phi$ is surjective, and hence an isometric isomorphism. $\qquad \square$

Before we prove Theorem 2.9, we state the following definition and theorems related to measurable functions for Banach-space valued functions.

**Definition C.2** ([12, p.4, Definition 5]). A function $H : \Omega \to \mathcal{H}$ is called an $\mathcal{F}$-simple function if it has the form $H = \sum_{i=1}^{n} h_i \mathbf{1}_{B_i}$ for some $h_i \in \mathcal{H}$ and $B_i \in \mathcal{F}$.

A function $H : \Omega \to \mathcal{H}$ is said to be $\mathcal{F}$-measurable if there is a sequence $(H_n)$ of $\mathcal{H}$-valued, $\mathcal{F}$-simple functions such that $H_n \to H$ pointwise.

**Theorem C.3** ([12, p.4, Theorem 6]). *If $H : \Omega \to \mathcal{H}$ is $\mathcal{F}$-measurable, then there is a sequence $(H_n)$ of $\mathcal{H}$-valued, $\mathcal{F}$-simple functions such that $H_n \to H$ pointwise and $|H_n| \leq |H|$ for every $n$.*

**Theorem C.4** ([12, p.19, Theorem 48], Lebesgue Convergence Theorem). *Let $(H_n)$ be a sequence in $L^1_{\mathcal{H}}(P)$, $H : \Omega \to \mathcal{H}$ a P-measurable function, and $g \in L^1_+(P)$ such that $H_n \to H$ P-almost everywhere and $|H_n| \leq g$, P-almost everywhere, for each $n$. Then $H \in L^1_{\mathcal{H}}(P)$ and $H_n \to H$ in $L^1_{\mathcal{H}}(P)$, i.e. $\int_\Omega H_n dP \to \int_\Omega H dP$.*

**Theorem 2.9.** Suppose that $P(\cdot \mid \mathcal{E})$ admits a regular version $Q$. Then $QH : \Omega \to \mathcal{H}$ with $\omega \mapsto Q_\omega H = \int_\Omega H(\omega') Q_\omega(d\omega')$ is a version of $\mathbb{E}[H \mid \mathcal{E}]$ for every Bochner P-integrable $H$.

*Proof.* Suppose $H$ is Bochner $P$-integrable. Since $Q$ is a regular version of $P(\cdot \mid \mathcal{E})$, it is a probability transition kernel from $(\Omega, \mathcal{E})$ to $(\Omega, \mathcal{F})$.

We first show that $QH$ is measurable with respect to $\mathcal{E}$. The map $Q : \Omega \to \mathcal{H}$ is well-defined, since, for each $\omega \in \Omega$, $Q_\omega H$ is the Bochner-integral of $H$ with respect to the measure $B \to Q_\omega(B)$. Since $H$ is $\mathcal{F}$-measurable, by Theorem C.3, there is a sequence $(H_n)$ of $\mathcal{H}$-valued, $\mathcal{F}$-simple functions such that $H_n \to H$ pointwise. Then for each $\omega \in \Omega$, $Q_\omega H = \lim_{n\to\infty} Q_\omega H_n$ by Theorem C.4. But for each $n$, we can write $H_n = \sum_{j=1}^{m} h_j \mathbf{1}_{B_j}$ for some $h_j \in \mathcal{H}$ and $B_j \in \mathcal{F}$, and so $Q_\omega H_n = \sum_{j=1}^{m} h_j Q_\omega(B_j)$. For each $B_j$ the map $\omega \mapsto Q_\omega(B_j)$ is $\mathcal{E}$-measurable (by the definition of transition probability kernel, Definition 2.7), and so as a linear combination of $\mathcal{E}$-measurable functions, $QH_n$ is $\mathcal{E}$-measurable. Hence, as a pointwise limit of $\mathcal{E}$-measurable functions, $QH$ is also $\mathcal{E}$-measurable, by [12, p.6, Theorem 10].

Next, we show that, for all $A \in \mathcal{E}$, $\int_A H dP = \int_A QH dP$. Fix $A \in \mathcal{E}$. By Theorem C.3, there is a sequence $(H_n)$ of $\mathcal{H}$-valued, $\mathcal{F}$-simple functions such that $H_n \to H$ pointwise. For each $n$, we can write $H_n = \sum_{j=1}^{m} h_j \mathbf{1}_{B_j}$ for some $h_j \in \mathcal{H}$ and $B_j \in \mathcal{F}$, and

$$
\begin{aligned}
\int_A QH_n dP &= \int_A \sum_{j=1}^{m} h_j Q(B_j) dP \\
&= \int_A \sum_{j=1}^{m} h_j P(B_j \mid \mathcal{E}) dP \quad \text{since } Q \text{ is a version of } P(\cdot \mid \mathcal{E}) \\
&= \sum_{j=1}^{m} h_j \int_A \mathbb{E}[\mathbf{1}_{B_j} \mid \mathcal{E}] dP \quad \text{by the definition of conditional probability measures} \\
&= \int_A \sum_{j=1}^{m} h_j \mathbf{1}_{B_j} dP \qquad \text{by the definition of conditional expectations, since } A \in \mathcal{E} \\
&= \int_A H_n dP.
\end{aligned}
$$

We have $H_n \to H$ pointwise by assertion, and as before, $QH_n \to QH$ pointwise. Hence,

$$
\begin{aligned}
\int_A QH dP &= \lim_{n\to\infty} \int_A QH_n dP \quad \text{by Theorem C.4} \\
&= \lim_{n\to\infty} \int_A H_n dP \qquad \text{by above} \\
&= \int_A H dP \qquad \text{by Theorem C.4.}
\end{aligned}
$$

Hence, by the definition of the conditional expectation, $QH$ is a version of $\mathbb{E}[H \mid \mathcal{E}]$. $\qquad\square$

**Lemma 3.2.** For any $f \in \mathcal{H}_{\mathcal{X}}$, $\mathbb{E}_{X|Z}[f(X) \mid Z] = \langle f, \mu_{P_{X|Z}} \rangle_{\mathcal{H}_{\mathcal{X}}}$ almost surely.

*Proof.* The left-hand side is the conditional expectation of the real-valued random variable $f(X)$ given $Z$. We need to check that the right-hand side is also that. Note that $\langle f, \mu_{P_{X|Z}} \rangle_{\mathcal{H}_{\mathcal{X}}}$ is clearly $Z$-measurable, and $P$-integrable (by the Cauchy-Schwarz inequality and the integrability condition (1)). Take any $A \in \sigma(Z)$. Then

$$
\begin{aligned}
\int_A \langle f, \mu_{P_{X|Z}} \rangle_{\mathcal{H}_{\mathcal{X}}} dP &= \int_A \left\langle f, \mathbb{E}_{X|Z}[k_{\mathcal{X}}(\cdot, X) \mid Z] \right\rangle_{\mathcal{H}_{\mathcal{X}}} dP && \text{by definition} \\
&= \left\langle f, \int_A \mathbb{E}_{X|Z}[k_{\mathcal{X}}(\cdot, X) \mid Z] dP \right\rangle_{\mathcal{H}_{\mathcal{X}}} && (+) \\
&= \left\langle f, \int_A k_{\mathcal{X}}(\cdot, X) dP \right\rangle_{\mathcal{H}_{\mathcal{X}}} && \text{see Definition 2.5} \\
&= \int_A \langle f, k_{\mathcal{X}}(\cdot, X) \rangle_{\mathcal{H}_{\mathcal{X}}} dP && (+) \\
&= \int_A f(X) dP && \text{by the reproducing property.}
\end{aligned}
$$

Here, in $(+)$, we used the fact that the order of a continuous linear operator and Bochner integration can be interchanged [12, p.30, Theorem 36]. Hence $\langle f, \mu_{P_{X|Z}} \rangle_{\mathcal{H}_{\mathcal{X}}}$ is a version of the conditional expectation $\mathbb{E}_{X|Z}[f(X) \mid Z]$. $\qquad\square$

**Lemma 3.3.** For any pair $f \in \mathcal{H}_{\mathcal{X}}$ and $g \in \mathcal{H}_{\mathcal{Y}}$, $\mathbb{E}_{XY|Z}[f(X)g(Y) \mid Z] = \langle f \otimes g, \mu_{P_{XY|Z}} \rangle_{\mathcal{H}_{\mathcal{X}} \otimes \mathcal{H}_{\mathcal{Y}}}$ almost surely.

*Proof.* The left-hand side is the conditional expectation of the real-valued random variable $f(X)g(Y)$ given $Z$. We need to check that the right-hand side is also that. Note that $\langle f \otimes g, \mu_{P_{XY|Z}} \rangle_{\mathcal{H}_{\mathcal{X}} \otimes \mathcal{H}_{\mathcal{Y}}}$ is clearly $Z$-measurable, and $P$-integrable (by the Cauchy-Schwarz inequality and the integrability condition (2)). Take any $A \in \sigma(Z)$. Then

$$
\begin{aligned}
\int_A \langle f \otimes g, \mu_{P_{XY|Z}} \rangle_{\mathcal{H}_{\mathcal{X}} \otimes \mathcal{H}_{\mathcal{Y}}} dP &= \int_A \left\langle f \otimes g, \mathbb{E}_{XY|Z}[k_{\mathcal{X}}(\cdot, X) \otimes k_{\mathcal{Y}}(\cdot, Y) \mid Z] \right\rangle_{\mathcal{H}_{\mathcal{X}} \otimes \mathcal{H}_{\mathcal{Y}}} dP \\
&= \left\langle f \otimes g, \int_A \mathbb{E}_{XY|Z}[k_{\mathcal{X}}(\cdot, X) \otimes k_{\mathcal{Y}}(\cdot, Y) \mid Z] dP \right\rangle_{\mathcal{H}_{\mathcal{X}} \otimes \mathcal{H}_{\mathcal{Y}}} \\
&= \left\langle f \otimes g, \int_A k_{\mathcal{X}}(\cdot, X) \otimes k_{\mathcal{Y}}(\cdot, Y) dP \right\rangle_{\mathcal{H}_{\mathcal{X}} \otimes \mathcal{H}_{\mathcal{Y}}} \\
&= \int_A \langle f \otimes g, k_{\mathcal{X}}(\cdot, X) \otimes k_{\mathcal{Y}}(\cdot, Y) \rangle_{\mathcal{H}_{\mathcal{X}} \otimes \mathcal{H}_{\mathcal{Y}}} dP \\
&= \int_A f(X) g(Y) dP.
\end{aligned}
$$

So $\langle f \otimes g, \mu_{P_{XY|Z}} \rangle_{\mathcal{H}_{\mathcal{X}} \otimes \mathcal{H}_{\mathcal{Y}}}$ is a version of the conditional expectation $\mathbb{E}_{XY|Z}[f(X)g(Y) \mid Z]$. $\quad\square$

**Theorem 4.1.** Assume that $\mathcal{H}_{\mathcal{X}}$ is separable, and denote its Borel $\sigma$-algebra by $\mathcal{B}(\mathcal{H}_{\mathcal{X}})$. Then we can write

$$\mu_{P_{X|Z}} = F_{P_{X|Z}} \circ Z,$$

where $F_{P_{X|Z}} : \mathcal{Z} \to \mathcal{H}_{\mathcal{X}}$ is some deterministic function, measurable with respect to $\mathfrak{Z}$ and $\mathcal{B}(\mathcal{H}_{\mathcal{X}})$.

*Proof.* Let $\mathrm{Im}(Z) \subseteq \mathcal{Z}$ be the image of $Z : \Omega \to \mathcal{Z}$, and let $\tilde{\mathfrak{Z}}$ denote the $\sigma$-algebra on $\mathrm{Im}(Z)$ defined by $\tilde{\mathfrak{Z}} = \{A \cap \mathrm{Im}(Z) : A \in \mathfrak{Z}\}$ (see [9, page 5, 1.15]). We will first construct a function $\tilde{F} : \mathrm{Im}(Z) \to \mathcal{H}_{\mathcal{X}}$, measurable with respect to $\tilde{\mathfrak{Z}}$ and $\mathcal{B}(\mathcal{H}_{\mathcal{X}})$, such that $\mu_{P_{X|Z}} = \tilde{F} \circ Z$.

For a given $z \in \mathrm{Im}(Z) \subseteq \mathcal{Z}$, we have $Z^{-1}(z) \subseteq \Omega$. Suppose for contradiction that there are two distinct elements $\omega_1, \omega_2 \in Z^{-1}(z)$ such that $\mu_{P_{X|Z}}(\omega_1) \neq \mu_{P_{X|Z}}(\omega_2)$. Since $\mathcal{H}_{\mathcal{X}}$ is Hausdorff,

there are disjoint open neighbourhoods $N_1$ and $N_2$ of $\mu_{P_{X|Z}}(\omega_1)$ and $\mu_{P_{X|Z}}(\omega_2)$ respectively. By definition of a Borel $\sigma$-algebra, we have $N_1, N_2 \in \mathcal{B}(\mathcal{H}_\mathcal{X})$, and since $\mu_{P_{X|Z}}$ is $\sigma(Z)$-measurable,

$$\mu_{P_{X|Z}}^{-1}(N_1), \mu_{P_{X|Z}}^{-1}(N_2) \in \sigma(Z). \tag{6}$$

Furthermore, $\mu_{P_{X|Z}}^{-1}(N_1)$ and $\mu_{P_{X|Z}}^{-1}(N_2)$ are neighbourhoods of $\omega_1$ and $\omega_2$ respectively, and are disjoint.

(i) For any $B \in \tilde{\mathfrak{Z}}$ with $z \in B$, since $Z(\omega_1) = z = Z(\omega_2)$, we have $\omega_1, \omega_2 \in Z^{-1}(B)$. So $Z^{-1}(B) \neq \mu_{P_{X|Z}}^{-1}(N_1)$ and $Z^{-1}(B) \neq \mu_{P_{X|Z}}^{-1}(N_2)$, as $\omega_2 \notin \mu_{P_{X|Z}}^{-1}(N_1)$ and $\omega_1 \notin \mu_{P_{X|Z}}^{-1}(N_2)$.

(ii) For any $B \in \tilde{\mathfrak{Z}}$ with $z \notin B$, we have $\omega_1 \notin Z^{-1}(B)$ and $\omega_2 \notin Z^{-1}(B)$. So $Z^{-1}(B) \neq \mu_{P_{X|Z}}^{-1}(N_1)$ and $Z^{-1}(B) \neq \mu_{P_{X|Z}}^{-1}(N_2)$.

Since $\sigma(Z) = \{Z^{-1}(B) \mid B \in \tilde{\mathfrak{Z}}\}$ (see [9], page 11, Exercise 2.20), we can't have $\mu_{P_{X|Z}}^{-1}(N_1) \in \sigma(Z)$ nor $\mu_{P_{X|Z}}^{-1}(N_2) \in \sigma(Z)$. This is a contradiction to (6). We therefore conclude that, for any $z \in \mathcal{Z}$, if $Z(\omega_1) = z = Z(\omega_2)$ for distinct $\omega_1, \omega_2 \in \Omega$, then $\mu_{P_{X|Z}}(\omega_1) = \mu_{P_{X|Z}}(\omega_2)$.

We define $\tilde{F}(z)$ to be the unique value of $\mu_{P_{X|Z}}(\omega)$ for all $\omega \in Z^{-1}(z)$. Then for any $\omega \in \Omega$, $\mu_{P_{X|Z}}(\omega) = \tilde{F}(Z(\omega))$ by construction. It remains to check that $\tilde{F}$ is measurable with respect to $\tilde{\mathfrak{Z}}$ and $\mathcal{B}(\mathcal{H}_\mathcal{X})$.

Take any $N \in \mathcal{B}(\mathcal{H}_\mathcal{X})$. Since $\mu_{P_{X|Z}}$ is $\sigma(Z)$-measurable, $\mu_{P_{X|Z}}^{-1}(N) = Z^{-1}(\tilde{F}^{-1}(N)) \in \sigma(Z)$. Since $\sigma(Z) = \{Z^{-1}(B) \mid B \in \tilde{\mathfrak{Z}}\}$, we have $Z^{-1}(\tilde{F}^{-1}(N)) = Z^{-1}(C)$ for some $C \in \tilde{\mathfrak{Z}}$. Since the mapping $Z : \Omega \to \text{Im}(Z)$ is surjective, $\tilde{F}^{-1}(N) = C$. Hence $\tilde{F}^{-1}(N) \in \tilde{\mathfrak{Z}}$, and so $\tilde{F}$ is measurable with respect to $\tilde{\mathfrak{Z}}$ and $\mathcal{B}(\mathcal{H}_\mathcal{X})$.

Finally, we can extend $\tilde{F} : \text{Im}(Z) \to \mathcal{H}_\mathcal{X}$ to $F : \mathcal{Z} \to \mathcal{H}_\mathcal{X}$ by [13, page 128, Corollary 4.2.7] (note that $\mathcal{H}_\mathcal{X}$ is a complete metric space, and assumed to be separable in this theorem). $\qquad \square$

**Theorem 4.2.** $F_{P_{X|Z}} \in L^2(\mathcal{Z}, P_Z; \mathcal{H}_\mathcal{X})$ minimises both $\tilde{\mathcal{E}}_{X|Z}$ and $\mathcal{E}_{X|Z}$, i.e.

$$F_{P_{X|Z}} = \underset{F \in L^2(\mathcal{Z}, P_Z; \mathcal{H}_\mathcal{X})}{\arg\min} \mathcal{E}_{X|Z}(F) = \underset{F \in L^2(\mathcal{Z}, P_Z; \mathcal{H}_\mathcal{X})}{\arg\min} \tilde{\mathcal{E}}_{X|Z}(F).$$

Moreover, it is almost surely unique, i.e. it is almost surely equal to any other minimiser of the objective functionals.

*Proof.* Recall that we have

$$\mathcal{E}_{X|Z}(F) := \mathbb{E}_Z \left[ \|F_{P_{X|Z}}(Z) - F(Z)\|_{\mathcal{H}_\mathcal{X}}^2 \right].$$

So clearly, $\mathcal{E}_{X|Z}(F_{P_{X|Z}}) = 0$, meaning $F_{P_{X|Z}}$ minimises $\mathcal{E}_{X|Z}$ in $L^2(\mathcal{Z}, P_Z; \mathcal{H}_\mathcal{X})$. So it only remains to show that $\tilde{\mathcal{E}}_{X|Z}$ is minimised in $L^2(\mathcal{Z}, P_Z; \mathcal{H}_\mathcal{X})$ by $F_{P_{X|Z}}$.

Let $F$ be any element in $L^2(\mathcal{Z}, P_Z; \mathcal{H}_\mathcal{X})$. Then we have

$$\begin{aligned}
\tilde{\mathcal{E}}_{X|Z}(F) - \tilde{\mathcal{E}}_{X|Z}(F_{P_{X|Z}}) &= \mathbb{E}_{X,Z}[\|k_\mathcal{X}(X, \cdot) - F(Z)\|_{\mathcal{H}_\mathcal{X}}^2] - \mathbb{E}_{X,Z}[\|k_\mathcal{X}(X, \cdot) - F_{P_{X|Z}}(Z)\|_{\mathcal{H}_\mathcal{X}}^2] \\
&= \mathbb{E}_Z[\|F(Z)\|_{\mathcal{H}_\mathcal{X}}^2] - 2\mathbb{E}_{X,Z}[\langle k_\mathcal{X}(X, \cdot), F(Z)\rangle_{\mathcal{H}_\mathcal{X}}] \\
&\quad + 2\mathbb{E}_{X,Z}\left[\langle k_\mathcal{X}(X, \cdot), F_{P_{X|Z}}(Z)\rangle_{\mathcal{H}_\mathcal{X}}\right] - \mathbb{E}_Z\left[\|F_{P_{X|Z}}(Z)\|_{\mathcal{H}_\mathcal{X}}^2\right].
\end{aligned} \tag{7}$$

Here,

$$\mathbb{E}_{X,Z}\left[\langle k_\mathcal{X}(X, \cdot), F(Z)\rangle_{\mathcal{H}_\mathcal{X}}\right] = \mathbb{E}_Z\left[\mathbb{E}_{X|Z}\left[F(Z)(X) \mid Z\right]\right] \qquad \text{by the reproducing property}$$

$$= \mathbb{E}_Z\left[\langle F(Z), \mu_{P_{X|Z}}\rangle_{\mathcal{H}_{\mathcal{X}}}\right] \qquad \text{by Lemma 3.2}$$

$$= \mathbb{E}_Z\left[\langle F(Z), F_{P_{X|Z}}(Z)\rangle_{\mathcal{H}_{\mathcal{X}}}\right] \qquad \text{since } \mu_{P_{X|Z}} = F_{P_{X|Z}} \circ Z$$

and similarly,

$$\mathbb{E}_{X,Z}[\langle k_{\mathcal{X}}(X, \cdot), F_{P_{X|Z}}(Z)\rangle_{\mathcal{H}_{\mathcal{X}}}] = \mathbb{E}_Z[\mathbb{E}_{X|Z}[F_{P_{X|Z}}(Z)(X) \mid Z]] \qquad \text{by the reproducing property}$$

$$= \mathbb{E}_Z\left[\langle F_{P_{X|Z}}(Z), F_{P_{X|Z}}(Z)\rangle_{\mathcal{H}_{\mathcal{X}}}\right] \quad \text{by Lemma 3.2}$$

$$= \mathbb{E}_Z\left[\|F_{P_{X|Z}}(Z)\|_{\mathcal{H}_{\mathcal{X}}}^2\right].$$

Substituting these expressions back into (7), we have

$$\tilde{\mathcal{E}}_{X|Z}(F) - \tilde{\mathcal{E}}_{X|Z}(F_{P_{X|Z}})$$
$$= \mathbb{E}_Z[\|F(Z)\|_{\mathcal{H}_{\mathcal{X}}}^2] - 2\mathbb{E}_Z[\langle F(Z), F_{P_{X|Z}}(Z)\rangle_{\mathcal{H}_{\mathcal{X}}}] + \mathbb{E}_Z[\|F_{P_{X|Z}}(Z)\|_{\mathcal{H}_{\mathcal{X}}}^2]$$
$$= \mathbb{E}_Z[\|F(Z) - F_{P_{X|Z}}(Z)\|_{\mathcal{H}_{\mathcal{X}}}^2]$$
$$\geq 0.$$

Hence, $F_{P_{X|Z}}$ minimises $\tilde{\mathcal{E}}_{X|Z}$ in $L^2(\mathcal{Z}, P_Z; \mathcal{H}_{\mathcal{X}})$. The minimiser is further more $P_Z$-almost surely unique; indeed, if $F' \in L^2(\mathcal{Z}, P_Z; \mathcal{H}_{\mathcal{X}})$ is another minimiser of $\tilde{\mathcal{E}}_{X|Z}$, then the calculation in (7) shows that

$$\mathbb{E}_Z\left[\|F_{P_{X|Z}}(Z) - F'(Z)\|_{\mathcal{H}_{\mathcal{X}}}^2\right] = 0,$$

which immediately implies that $\|F_{P_{X|Z}}(Z) - F'(Z)\|_{\mathcal{H}_{\mathcal{X}}} = 0$ $P_Z$-almost surely, which in turn implies that $F_{P_{X|Z}} = F'$ $P_Z$-almost surely. $\qquad\square$

**Theorem 4.4.** Suppose that $k_{\mathcal{X}}$ and $k_{\mathcal{Z}}$ are bounded kernels, i.e. there exist $B_{\mathcal{Z}}, B_{\mathcal{X}} > 0$ such that $\sup_{z \in \mathcal{Z}} k_{\mathcal{Z}}(z, z) \leq B_{\mathcal{Z}}$ and $\sup_{x \in \mathcal{X}} k_{\mathcal{X}}(x, x) \leq B_{\mathcal{X}}$, and that the operator-valued kernel $l_{\mathcal{X}\mathcal{Z}}$ is $\mathcal{C}_0$-universal. Let the regularisation parameter $\lambda_n$ decay to 0 at a slower rate than $\mathcal{O}(n^{-1/2})$. Then our learning algorithm that produces $\hat{F}_{P_{X|Z},n,\lambda_n}$ is universally consistent (in the surrogate loss $\tilde{\mathcal{E}}_{X|Z}$), i.e. for any joint distribution $P_{XZ}$ and constants $\epsilon > 0$ and $\delta > 0$,

$$P_{XZ}(\tilde{\mathcal{E}}_{X|Z}(\hat{F}_{P_{X|Z},n,\lambda_n}) - \tilde{\mathcal{E}}_{X|Z}(F_{P_{X|Z}}) > \epsilon) < \delta$$

for large enough $n$.

*Proof.* Follows immediately from [37, Theorem 2.3]. 

$\qquad\square$

**Theorem 4.5.** In addition to the setting in Theorem 4.4, assume that $F_{P_{X|Z}} \in \mathcal{G}_{\mathcal{X}\mathcal{Z}}$. Let the regularisation parameter $\lambda_n$ decay to 0 with rate $\mathcal{O}(n^{-1/4})$. Then $\tilde{\mathcal{E}}_{X|Z}(\hat{F}_{P_{X|Z},n,\lambda_n}) - \tilde{\mathcal{E}}_{X|Z}(F_{P_{X|Z}}) = \mathcal{O}_P(n^{-1/4})$.

*Proof.* Follows immediately from [37, Theorem 2.4]. $\qquad\square$

**Theorem 5.2.** Suppose that $k_{\mathcal{X}}$ is a characteristic kernel, that $P_Z$ and $P_{Z'}$ are absolutely continuous with respect to each other, and that $P(\cdot \mid Z)$ and $P(\cdot \mid Z')$ admit regular versions. Then $\text{MCMD}_{P_{X|Z}, P_{X'|Z'}} = 0$ $P_Z$- (or $P_{Z'}$-)almost everywhere if and only if, for $P_Z$- (or $P_{Z'}$-)almost all $z \in \mathcal{Z}$, $P_{X|Z=z}(B) = P_{X'|Z'=z}(B)$ for all $B \in \mathfrak{X}$.

*Proof.* Write $Q$ and $Q'$ for some regular versions of $P(\cdot \mid Z)$ and $P(\cdot \mid Z')$ respectively, and assume without loss of generality that the conditional distributions $P_{X|Z}$ and $P_{X'|Z'}$ are given by $P_{X|Z}(\omega)(B) = Q_\omega(X \in B)$ and $P_{X'|Z'}(\omega)(B) = Q'_\omega(X' \in B)$ for $B \in \mathfrak{X}$. By the definition of regular versions, for each $B \in \mathfrak{X}$, the real-valued random variables $\omega \mapsto P_{X|Z}(\omega)(B)$ and $\omega \mapsto P_{X'|Z'}(\omega)(B)$ are measurable with respect to $Z$ and $Z'$ respectively, and so there are functions $R_B : \mathcal{Z} \to \mathbb{R}$ and $R'_B : \mathcal{Z} \to \mathbb{R}$ such that $P_{X|Z}(\omega)(B) = R_B(Z(\omega))$ and $P_{X'|Z'}(\omega)(B) =$

$R'_B(Z'(\omega))$. Moreover, for each fixed $z \in \mathcal{Z}$, the mappings $B \mapsto P_{X|Z}(Z^{-1}(z))(B) = R_B(z)$ and $B \mapsto P_{X'|Z'}(Z'^{-1}(z))(B) = R'_B(z)$ are measures. We write $R_B(z) = P_{X|Z=z}(B)$ and $R'_B(z) = P_{X'|Z'=z}(B)$.

By Theorem 2.9, there exists an event $A_1 \in \mathcal{F}$ with $P(A_1) = 1$ such that for all $\omega \in A_1$,

$$\mu_{P_{X|Z}}(\omega) := \mathbb{E}_{X|Z}[k_{\mathcal{X}}(X, \cdot) \mid Z](\omega) = \int_\Omega k_{\mathcal{X}}(X(\omega'), \cdot)Q_\omega(d\omega') = \int_{\mathcal{X}} k_{\mathcal{X}}(x, \cdot)P_{X|Z}(\omega)(dx),$$

and an event $A_2 \in \mathcal{F}$ with $P(A_2) = 1$ such that for all $\omega \in A_2$,

$$\mu_{P_{X'|Z'}}(\omega) := \mathbb{E}_{X'|Z'}[k_{\mathcal{X}}(X', \cdot) \mid Z'](\omega) = \int_\Omega k_{\mathcal{X}}(X'(\omega'), \cdot)Q_\omega(d\omega')$$
$$= \int_{\mathcal{X}} k_{\mathcal{X}}(x', \cdot)P_{X'|Z'}(\omega)(dx').$$

Suppose for contradiction that there exists some $D \in \mathfrak{Z}$ with $P_Z(D) > 0$ such that for all $z \in D$, $F_{P_{X|Z}}(z) \neq \int_{\mathcal{X}} k_{\mathcal{X}}(x, \cdot)R_{dx}(z)$. Then $P(Z^{-1}(D)) = P_Z(D) > 0$, and hence $P(Z^{-1}(D) \cap A_1) > 0$. For all $\omega \in Z^{-1}(D) \cap A_1$, we have $Z(\omega) \in D$, and hence

$$\mu_{P_{X|Z}}(\omega) = F_{P_{X|Z}}(Z(\omega)) \neq \int_{\mathcal{X}} k_{\mathcal{X}}(x, \cdot)R_{dx}(Z(\omega)) = \int_{\mathcal{X}} k_{\mathcal{X}}(x, \cdot)P_{X|Z}(\omega)(dx).$$

This contradicts our assertion that $\mu_{P_{X|Z}}(\omega) = \int_{\mathcal{X}} k_{\mathcal{X}}(x, \cdot)P_{X|Z}(\omega)(dx)$ for all $\omega \in A_1$, hence there does not exist $D \in \mathfrak{Z}$ with $P_Z(D) > 0$ such that for all $z \in D$, $F_{P_{X|Z}}(z) \neq \int_{\mathcal{X}} k_{\mathcal{X}}(x, \cdot)R_{dx}(z)$. Therefore, there must exist some $C_1 \in \mathfrak{Z}$ with $P_Z(C_1) = 1$ such that for all $z \in C_1$, $F_{P_{X|Z}}(z) = \int_{\mathcal{X}} k_{\mathcal{X}}(x, \cdot)R_{dx}(z)$. Similarly, there must exist some $C_2 \in \mathfrak{Z}$ with $P_Z(C_2) = 1$ such that for all $z \in C_2$, $F_{P_{X'|Z'}}(z) = \int_{\mathcal{X}} k_{\mathcal{X}}(x, \cdot)R'_{dx}(z)$. Since $P_Z$ and $P_{Z'}$ are absolutely continuous with respect to each other, we also have $P_Z(C_2) = 1 = P_{Z'}(C_1)$.

($\Longrightarrow$) Suppose first that $\mathrm{MCMD}_{P_{X|Z}, P_{X'|Z'}} = \|F_{P_{X|Z}} - F_{P_{X'|Z'}}\|_{\mathcal{H}_{\mathcal{X}}} = 0$ $P_Z$-almost everywhere, i.e. there exists $C \in \mathfrak{Z}$ with $P_Z(C) = 1$ such that for all $z \in C$, $\|F_{P_{X|Z}}(z) - F_{P_{X'|Z'}}(z)\|_{\mathcal{H}_{\mathcal{X}}} = 0$. Then for each $z \in C \cap C_1 \cap C_2$,

$$\int_{\mathcal{X}} k_{\mathcal{X}}(x, \cdot)R_{dx}(z) = F_{P_{X|Z}}(z) \qquad \text{since } z \in C_1$$
$$= F_{P_{X'|Z'}}(z) \qquad \text{since } z \in C$$
$$= \int_{\mathcal{X}} k_{\mathcal{X}}(x, \cdot)R'_{dx}(z) \qquad \text{since } z \in C_2.$$

Since the kernel $k_{\mathcal{X}}$ is characteristic, this means that $B \mapsto R_B(z)$ and $B \mapsto R'_B(z)$ are the same probability measure on $(\mathcal{X}, \mathfrak{X})$. By countable intersection, we have $P_Z(C \cap C_1 \cap C_2) = 1$, so $P_Z$-almost everywhere,

$$P_{X|Z=z}(B) = P_{X'|Z'=z}(B)$$

for all $B \in \mathfrak{X}$.

($\Longleftarrow$) Now assume there exists $C \in \mathfrak{Z}$ with $P_Z(C) = 1$ such that for each $z \in C$, $R_B(z) = R'_B(z)$ for all $B \in \mathfrak{X}$. Then for all $z \in C \cap C_1 \cap C_2$,

$$\left\|F_{P_{X|Z}}(z) - F_{P_{X'|Z'}}(z)\right\|_{\mathcal{H}_{\mathcal{X}}}$$
$$= \left\|\int_{\mathcal{X}} k_{\mathcal{X}}(x, \cdot)R_{dx}(z) - \int_{\mathcal{X}} k_{\mathcal{X}}(x, \cdot)R'_{dx}(z)\right\|_{\mathcal{H}_{\mathcal{X}}} \qquad \text{since } z \in C_1 \cap C_2$$
$$= \left\|\int_{\mathcal{X}} k_{\mathcal{X}}(x, \cdot)R_{dx}(z) - \int_{\mathcal{X}} k_{\mathcal{X}}(x, \cdot)R_{dx}(z)\right\|_{\mathcal{H}_{\mathcal{X}}} \qquad \text{since } z \in C$$
$$= 0,$$

and since $P_Z(C \cap C_1 \cap C_2) = 1$, $\|F_{P_{X|Z}} - F_{P_{X'|Z'}}\|_{\mathcal{H}_{\mathcal{X}}} = 0$ $P_Z$-almost everywhere.

$\square$

**Theorem 5.4.** Suppose $k_{\mathcal{X}} \otimes k_{\mathcal{Y}}$ is a characteristic kernel on $\mathcal{X} \times \mathcal{Y}$, and that $P(\cdot \mid Z)$ admits a regular version. Then $\mathrm{HSCIC}(X, Y \mid Z) = 0$ almost surely if and only if $X \perp\!\!\!\perp Y \mid Z$.

*Proof.* Write $Q$ for a regular version of $P(\cdot \mid Z)$, and assume without loss of generality that the conditional distributions $P_{X|Z}$, $P_{Y|Z}$ and $P_{XY|Z}$ are given by $P_{X|Z}(\omega)(B) = Q_\omega(X \in B)$ for $B \in \mathcal{X}$, $P_{Y|Z}(\omega)(C) = Q_\omega(Y \in C)$ for $C \in \mathfrak{Y}$ and $P_{XY|Z}(\omega)(D) = Q_\omega((X, Y) \in D)$ for $D \in \mathfrak{X} \times \mathfrak{Y}$. By Theorem 2.9, there exists an event $A_1 \in \mathcal{F}$ with $P(A_1) = 1$ such that for all $\omega \in A_1$,

$$\mu_{P_{X|Z}}(\omega) := \mathbb{E}_{X|Z}[k_{\mathcal{X}}(X, \cdot) \mid Z](\omega) = \int_\Omega k_{\mathcal{X}}(X(\omega'), \cdot) Q_\omega(d\omega') = \int_{\mathcal{X}} k_{\mathcal{X}}(x, \cdot) P_{X|Z}(\omega)(dx),$$

an event $A_2 \in \mathcal{F}$ with $P(A_2) = 1$ such that for all $\omega \in A_2$,

$$\mu_{P_{Y|Z}}(\omega) := \mathbb{E}_{Y|Z}[k_{\mathcal{Y}}(Y, \cdot) \mid Z](\omega) = \int_\Omega k_{\mathcal{Y}}(Y(\omega'), \cdot) Q_\omega(d\omega') = \int_{\mathcal{Y}} k_{\mathcal{Y}}(y, \cdot) P_{Y|Z}(\omega)(dy),$$

and an event $A_3 \in \mathcal{F}$ with $P(A_3) = 1$ such that for all $\omega \in A_3$,

$$\mu_{P_{XY|Z}}(\omega) = \int_{\mathcal{X} \times \mathcal{Y}} k_{\mathcal{X}}(x, \cdot) \otimes k_{\mathcal{Y}}(y, \cdot) P_{XY|Z}(\omega)(d(x, y)).$$

This means that, for each $\omega \in A_1$, $\mu_{P_{X|Z}}(\omega)$ is the mean embedding of $P_{X|Z}(\omega)$, and for each $\omega \in A_2$, $\mu_{P_{Y|Z}}(\omega)$ is the mean embedding of $P_{Y|Z}(\omega)$.

$(\implies)$ Suppose first that $\mathrm{HSCIC}(X, Y \mid Z) = \|\mu_{P_{XY|Z}} - \mu_{P_{X|Z}} \otimes \mu_{P_{Y|Z}}\|_{\mathcal{H}_{\mathcal{X}} \otimes \mathcal{H}_{\mathcal{Y}}} = 0$ almost surely, i.e. there exists $A \in \mathcal{F}$ with $P(A) = 1$ such that for all $\omega \in A$, $\|\mu_{P_{XY|Z}}(\omega) - \mu_{P_{X|Z}}(\omega) \otimes \mu_{P_{Y|Z}}(\omega)\|_{\mathcal{H}_{\mathcal{X}} \otimes \mathcal{H}_{\mathcal{Y}}} = 0$. Then for each $\omega \in A \cap A_1 \cap A_2 \cap A_3$,

$$\begin{aligned}
\int_{\mathcal{X} \times \mathcal{Y}} k_{\mathcal{X}}(x, \cdot) \otimes k_{\mathcal{Y}}(y, \cdot) P_{XY|Z}(\omega)(d(x, y)) &= \mu_{P_{XY|Z}}(\omega) && \text{since } \omega \in A_3 \\
&= \mu_{P_{X|Z}}(\omega) \otimes \mu_{P_{Y|Z}}(\omega) && \text{since } \omega \in A \\
&= \int_{\mathcal{X}} k_{\mathcal{X}}(x, \cdot) P_{X|Z}(\omega)(dx) \otimes \int_{\mathcal{Y}} k_{\mathcal{Y}}(y, \cdot) P_{Y|Z}(\omega)(dy) && \text{since } \omega \in A_1 \cap A_2 \\
&= \int_{\mathcal{X} \times \mathcal{Y}} k_{\mathcal{X}}(x, \cdot) \otimes k_{\mathcal{Y}}(y, \cdot) P_{X|Z}(\omega) P_{Y|Z}(\omega)(d(x, y)) && \text{by Fubini.}
\end{aligned}$$

Since the kernel $k_{\mathcal{X}} \otimes k_{\mathcal{Y}}$ is characteristic, the distributions $P_{XY|Z}(\omega)$ and $P_{X|Z}(\omega) P_{Y|Z}(\omega)$ on $\mathcal{X} \times \mathcal{Y}$ are the same. By countable intersection, we have $P(A \cap A_1 \cap A_2 \cap A_3) = 1$, so $P_{XY|Z}$ and $P_{X|Z} P_{Y|Z}$ are the same almost surely, and we have $X \perp\!\!\!\perp Y \mid Z$.

$(\impliedby)$ Now assume $X \perp\!\!\!\perp Y \mid Z$, i.e. there exists $A \in \mathcal{F}$ with $P(A) = 1$ such that for each $\omega \in A$, the distributions $P_{XY|Z}(\omega)$ and $P_{X|Z}(\omega) P_{Y|Z}(\omega)$ are the same. Then for all $\omega \in A \cap A_1 \cap A_2 \cap A_3$,

$$\begin{aligned}
\mu_{P_{XY|Z}}(\omega) &= \int_{\mathcal{X} \times \mathcal{Y}} k_{\mathcal{X}}(x, \cdot) \otimes k_{\mathcal{Y}}(y, \cdot) P_{XY|Z}(\omega)(d(x, y)) && \text{since } \omega \in A_3 \\
&= \int_{\mathcal{X} \times \mathcal{Y}} k_{\mathcal{X}}(x, \cdot) \otimes k_{\mathcal{Y}}(y, \cdot) P_{X|Z}(\omega)(dx) P_{Y|Z}(\omega)(dy) && \text{since } \omega \in A \\
&= \int_{\mathcal{X}} k_{\mathcal{X}}(x, \cdot) P_{X|Z}(\omega)(dx) \otimes \int_{\mathcal{Y}} k_{\mathcal{Y}}(y, \cdot) P_{Y|Z}(\omega)(dy) && \text{by Fubini} \\
&= \mu_{P_{X|Z}}(\omega) \otimes \mu_{P_{Y|Z}}(\omega) && \text{since } \omega \in A_1 \cap A_2.
\end{aligned}$$

and since $P(A \cap A_1 \cap A_2 \cap A_3) = 1$, $\mathrm{HSCIC}(X, Y \mid Z) = 0$ almost surely.

$\square$