[Reviews · NeurIPS 2020]

Review 1

Summary and Contributions: Note: For the convenience of the authors in their response, I will label each of my points with [Section.Paragraph] (e.g. [1.2]). Authors, please also start your paragraphs with the same numbering so I know which of my comments you are referring to. The authors propose a measure-theoretic approach to formulating conditional mean embeddings, in contrast to the operator approach currently within the kernel mean embedding literature. The primary contribution comes from redefining the conditional mean embedding as the Bochner conditional expectation of the canonical feature of an RKHS - informally, leaving the conditioned event as a random variable. Consequently, they also redefine the MCMD (Maximum Conditional Mean Discrepancy) and HSCIC (Hilbert-Schmidt Conditional Independence Criterion) along the same lines. They also obtain empirical estimates of the CME from a vector-valued regression perspective, and plug this empirical estimate of the CME to obtain empirical estimates for the MCMD and HSCIC. Finally, they have provided improved convergence rates of CMEs from $O_{p}(n{-1/2})$ which is faster than $O_{p}((n \lambda)^{-1/2} + \lambda^{\1/2})$ in the current literature which is at best $O_{p}(n{-1/4})$ with the right regularization decay rate.

Strengths: [2.1] The paper's take on using a measure theoretic approach to redefine CMEs strikes me as very valuable to the kernel mean embedding community, even though in terms of novelty it is unsurprising. I would argue that this paper's main strengths is not in its novelty, but in that it has formalized an intuition that the kernel mean embedding community already had regarding the CME. The core contribution of the redefinition of the CME is essentially making the distinction between $\mu_{X | Z = \cdot}$ and $\mu_{X | Z}$, here using the notation more familiar to us in the literature. This is formalized through Bochner conditional expectations in the latter and explicitly treating $\mu_{X | Z}$ as a Z-measurable random variable with values in $\mathcal{H}_{\mathcal{X}}$. The results that follows from this formalization is technically sound, however I do not feel that this should be communicated as "a new operator-free, measure-theoretic approach" as in the Abstract. Nevertheless, similar to in functional analysis the realization that a function $f: \mathcal{X} \to \mathbb{R}$ can be informally thought of as a vector ${f(x)}_{x \in \mathcal{X}$ by enumerating its evaluations throughout all $x \mathcal{X}$, this sort of insight is still valuable. ==========After Rebuttal========== I'd like to thank the authors for the thorough response in their rebuttal. [3.1] I agree the reformulation is valuable in that it opens up new directions of research for CMEs. [3.3] Thanks for the high level explanation. [3.4] Provided that the experiments do verify the claims of Theorem 4.4, I am happy to revise my score to a 7. [3.5] Yes, please try to include this in the main paper, despite the space constraints. [6.3] I agree it is a more principled way of motivating regression. Thanks for clarifying. [5.3] Yes, please try to include this in the main paper, despite the space constraints. [5.4] Yes, a short mention in the main paper would be fine, but please add the details to support this in the appendix for completeness (although it should be unsurprising to most readers that the claims remain the same). Overall, my stance on the paper has improved and I have revised my score from a 6 to a 7, provided the authors honour what was promised for inclusion in the camera-ready version.

Weaknesses: [3.1] This paper is an example of applying one great shift of formulation to an existing literature. As I alluded above in the Strengths section, the core idea presented in this paper is to reformulate the literature around CMEs by treating all relevant and related objects as a Z-measurable random variable instead of conditioned on a single event {Z = z}. Those relevant objects in this paper are the CME itself, the MCMD, and the HSCIC. In this regard the novelty is not particularly strong despite the very comprehensive reformulation. On the empirical side, the concrete consequence of treating all relevant and related objects as a Z-measurable random variable instead of conditioned on a single event {Z = z} simply translated to having all the relevant quantities left as a function of $z \in \mathcal{Z}$, and the form of the empirical estimators remained identical to what we already have in the kernel mean embedding literature. The vector-valued regression view is not new either (Grunewalder et. al., 2012). As a result, this paper reads more as a very well thought remarks paper on the existing literature with a single novel change to its formulation. However, as I mentioned in the Strengths section, this is still nevertheless valuable, and my overall stance is more positive than negative, although these high level concerns would add to my hesitation in the final decision. [3.2] It is more appropriate to say that what you have attempted to redefine in Definition 3.1 is not the CME (conditional mean embedding) but the CMO (conditional mean operator). Only when the Bochner conditional expectation is realized for a particular {Z = z} event is it a conditional mean embedding. This would make it clearer how your proposed reformulation fits into the existing literature (Fukumizu et. al., 2004, Song et. al., 2009). At the same time, I can understand the desire to avoid this nomenclature here in your paper in order to emphasize that your definition does not start from an operator angle. Nevertheless, you may want to spend some time thinking about whether there is a better terminology that would not to give the broader audience the impression that this object you've redefined is an embedding of a single distribution, but an embedding of the conditional distribution as a function of the conditioned variable. For example, Song et. al. (2013) emphasized this visually in their Figure 5, but you have the added complexity of the need to also visualize or communicate the fact that the Bochner conditional expectation is a random variable. [3.3] Regarding Theorem 4.4, can you provide an intuition why your rate $O_{p}(n{-1/2})$ is faster than $O_{p}((n \lambda)^{-1/2} + \lambda^{\1/2})$ in the current literature which is at best $O_{p}(n{-1/4})$ with the right regularization decay rate? Admittedly I did not have the bandwidth to go through the entire proof, so I am just looking for a high level summary or intuition here. While I understand they are formulated from different angles (measure-theoretic and operator-based), your empirical estimates are the same as what is in the literature when they are evaluated and computed, so why is the convergence faster now? [3.4] Adding to the point above on Theorem 4.4, I think this rate is a significant contribution and any reader would anticipate for you to verify this in the experimental section. The paper would be significantly stronger if you can support your claim in Theorem 4.4 with some experiments - even some toy ones might do. [3.5] Also regarding Theorem 4.4, could you also provide the general convergence rate as a function of both $n$ and $\lambda$? Also, it would be best if you can communicate with the notation $O_{p}$ instead of $O$ on the inequality on the probability of large error. I understand it means the same thing - but this would help readers to directly and more easily compare and contrast your theorems with that of Song et. al. (2009 and 2013).

Correctness: [4.1] The claims and methods seem correct, although I have not dived deep into the proofs of each theorem. [4.2] In Page 2, Line 70: The isomorphism should be $\Phi : \mathcal{H}_{\mathcal{X}} \otimes \mathcal{H}_{\mathcal{Y}} \to HS(\mathcal{H}_{\mathcal{Y}}, \mathcal{H}_{\mathcal{X}})

Clarity: [5.1.] The paper is relatively well written. The authors paid special attention to subscripting the mean embeddings with the probability measure they embed rather than the random variables, which I appreciate. The notations are also relatively clear and consistent. [5.2] One suggestion is to not use $\Phi$ as the isomorphism - this is usually reserved for the feature matrix even if they are uncountably infinite in size. [5.3] The notion of a regular version is quite vital to your discussion in Section 3, so I do not agree with the decision to leave this as part of the appendix. Please move the definition and discussion of regular versions into section 3 and make it extremely clear to readers who may not be immediately familiar with it. [5.4] In Page 2, Line 69: You used centered cross-covariance operators throughout without particular mention of why. The properties and results you used also hold for uncentered cross-covariance operators. Perhaps it would be good to clarify this choice and whether there is any particular reason against formulating the framework with uncentered cross-covariance operators.

Relation to Prior Work: [6.1] The paper makes an effort to emphasize the difference in its approach compared to the existing literature as this difference is its core contribution. In particular, they provided the common definition of CMOs and CMEs from Song et. al. (2009) in Definition 3.4 which contrasts with their definition in Definition 3.1. The main difference is that the authors defined CMEs using Bochner conditional expectations, rather than constructing them from operators that are isometric to cross-covariance operators. [6.2] The authors also discussed the difference between their HSCIC formulation with that of Sheng and Sriperumbudur (2019). [6.3] Given that the empirical estimates of the CME result in the same form as that of the literature when evaluated and computed, I would like to see more of a discussion here on what exactly is the contribution compared to the literature for the empirical aspect. As mentioned before, the vector-valued regression perspective is not new (Grunewalder et. al., 2012) either so in terms of formulation it is still mainly the use of Bochner conditional expectations at the start.

Reproducibility: Yes

Additional Feedback:


Review 2

Summary and Contributions: This paper proposes a new definition for the conditional mean embedding of conditional distributions. Instead of being an operator, the new CME is a Hilbert valued random variable. New definitions for MMD and HSIC derive from this definition. Estimation of the proposed CME is made through the resolution of a regression problem in a vv-RKHS. Guarantees are proposed through a surrogate loss

Strengths: - the paper is clear and well-written - the ideas are natural and well exposed/motivated - mathematical aspects are thoroughly dealt with - comparison with previous works are duly treated - the estimation/regression approach makes use of nice ideas to establish the guarantees

Weaknesses: - experiments could be strengthened, as they only provide a "proof of concept". However I find the theoretical contribution interesting enough for this aspect not to be too penalizing - recent works have extended the regression framework of vv-RKHS in infinite dimension to more losses than the square norm in the output space, and to other kernels than the identity decomposable (Duality in RKHSs with Infinite Dimensional Outputs: Application to Robust Losses, Laforgue et al. 2019). Do authors think it can be applied to the regression performed in Sec. 4, with possible advantages in terms of robustness? - have authors considered other kernels than the identity decomposable? What happens when the kernel is not C_0-universal, would an approximation scheme be possible? - why not considering the expected (w.r.t. Z) MCMD as discrepancy between conditional distributions? It seems that Thm 3.7 would still hold, while having a simpler (as not random) criterion. Also the latter could then be used to practice tests, can authors comment on this? - can the proposed approach be extended to the case where the conditioning variable is not the same? - I point out that original ideas on algorithmic stability come from "Stability and Generalization", Bousquet & Elisseeff 2002, the work by Kadri et al. 2016 being only a straightforward adaptation to vv-RKHSs (as opposed to scalar ones). In my opinion the original work would deserve citation

Correctness: Yes

Clarity: Yes

Relation to Prior Work: Yes

Reproducibility: Yes

Additional Feedback: Overall evaluation ********************* Despite limited numerical experiments, I find the theoretical contributions of the present paper of significant interest, well exposed, and thoroughly tackled, motivating my score. Post rebuttal *************** I have read other reviews and the response provided by the authors. As the modifications asked are not too critical in my opinion (provided that experiments indeed validate Thm 4.4), I keep my 7 score.


Review 3

Summary and Contributions: This paper presents a measure-theoretic approach for Kernel conditional mean embeddings. The work is a theoretical exercise aimed at improving the prior framework in which such notions and definitions of CME were previously provided.

Strengths: The paper presents a measure-theoretic setting for Kernel CMEs. It is mainly a theoretical exercise aimed at improving prior framework.

Weaknesses: The major weakness of this work is that it is meant only as a theoretical exercise providing variations to previously known definitions such as CME, conditional class covariance etc. A major emphasis is placed by the authors on the difference in definitions as compared to prior work. What is clearly missing is the practical relevance of this exercise.

Correctness: The claims and method seem correct. There are hardly any empirical results shown in the paper and what is shown is just a numerical illustration.

Clarity: The paper is not well written. One of the major problems I had reading this paper was that a lot of definitions are given in the supplementary material and in the main paper, the supplementary material is referred to for these definitions.

Relation to Prior Work: Differences from prior work in terms of definitions and assumptions are explained by the authors. What is not clear is how this exercise is useful.

Reproducibility: Yes

Additional Feedback: Post rebuttal: After the feedback from the authors, I am happy to reconsider my scores even though I am still not entirely convinced about the novelty of this work. I still think that it is a theoretical exercise and the authors haven't sufficiently demonstrated the practical relevance by a more real-life example. Applying a measure theoretic framework for the given problem is a good step but if we view the paper as a purely theoretical paper, the contribution is then primarily in the application of well known measure theory tools to a given setting. In that sense, in my view, the novelty is not high enough.


Review 4

Summary and Contributions: The authors pursue a measure-theoretic construction of conditional kernel mean embeddings which require less stringent assumptions than similar, previous results. The authors are able to provide constructions that do not rely on certain operators to have inverses, which is known to likely not be true in the first place. The authors then demonstrate how this framework can be used in the context of vector-valued RKHS regression, generating empirical estimates of the conditional kernel mean embeddings, and also provide sharper rates of convergence than previous analyses with similar assumptions.

Strengths: The main contributions of this work are: * It defines the conditional kernel mean embedding of a conditional distribution without relying on assumptions about operator invertibility. This extends to definitions of the MCMD and HSIC as well. * It provides an empirical estimate for the condition kernel mean embedding that converges to the true kernel mean embedding at a O_P(n^{-1/2}) rate and relies on less stringent assumptions, like output space H_X being finite dimensional. Thus this paper offers similar results to previous analyses, but with less strigent assumptions and sharper rates. The claims look sound (they are analogous to many of the results in the scalar-valued RKHS theory) and their empirical examples provide evidence of their methods generating sensible estimates and witness functions. Their HSIC experiment shows that their method of estimation produces relatively correct values for the various pairs of conditional distributions assessed.

Weaknesses: The biggest weakness of the work, which I am not sure is major, is determining how much weaker the necesary assumptions are by employing the conditional mean kernel embeddings from a measure-theoretic approach. The authors do a good job comparing their methods to other work and arguing the differences are substantial, but its not clear how often the previous stronger assumptions are violated.

Correctness: The claimed results in the paper are analagous to what one would expect for scalar-valued RKHSes and also from conditional distributions in measure theory. The estimators provided in the main text look correct, and while I haven't checked all the work in the appendix, the ideas there look apt for tackling the theorems cited in the main paper.

Clarity: Yes, the paper is well written. I personally would prefer if the authors avoiding using quotations when discussion the intuition behind their results (and instead just used simpler notation inspired from density functions like p(x|z), etc.), but that is my biggest claim with the clarity. I also was a bit confused at first when the authors kept referencing their "algorithm" but I didn't really see it defined anywhere.

Relation to Prior Work: Yes, the explanation given on L131-L156 was quite clear and specific. I appreciated the detail the authors provided to explain how their work differed from other similar analyses.

Reproducibility: Yes

Additional Feedback: L127: this is a bit confusing: the "if condition" already uses f. Supp info: L650: there is text going off the page I can't read. ==== Post Rebuttal ==== After reading the other reviewers' comments and the author rebuttal, I stand by my rating of 7.

[Author Response · NeurIPS 2020]

**We thank all reviewers for thorough and constructive feedback. Due to space constraints, we only discuss major**
**concerns here and will incorporate all of your concerns in our camera-ready version.**
\*\*When we refer to **Reviewer #n, m** within the text, we mean *our response to that point*, not the original review.\*\*

**Reviewer #1, [3.1]** We appreciate and are grateful for the overall positive stance on our submission. We agree that the
main contribution is to provide a shift of formulation of an existing concept with solid theoretical foundations, but
we believe novel ideas and applications can often stem from such new formulations that could not have been born
from the existing approaches (e.g. see **Reviewer #2, 4**). We thus believe that our contribution is well-aligned with the
NeurIPS evaluation criteria, which seek "[s]olid, technical papers that explore new territory or point out new directions
of research [...]". We also believe there are potential contributions made on the empirical side – please see **[6.3]** below.
**[3.3]** The key ideas in the proof are Chebyshev's inequality and the stability notion in [Bousquet and Elysseeff, 2002],
which are by now standard techniques in learning theory. The main difference is that operator-based approaches are not
based on vv-regression, but rather the theory of linear operators, which does not facilitate the use of such techniques.
**[3.4]** We propose to run some experiments to verify Theorem 4.4 and include the results in the camera-ready version.
**[3.5]** This is a valid point. The exact expression for the convergence rate are expressed on page 22/23 of the Appendix
as ($\dagger$), ($\dagger\dagger$) and ($\dagger\,\dagger\,\dagger$). We propose to include this as part of Theorem 4.4 in the camera-ready version.

**[6.3]** The contributions over [Grünewälder et al., 2012] (hereafter written [G]) are as follows. We apologise for not
making them clear in the submission and we propose to make more explicit in the camera-ready version. (i) We consider
the CME as an explicit function $\mathcal{Z} \to \mathcal{H}_{\mathcal{X}}$, which we believe is a more principled way of motivating regression, as
opposed to [G] who apply the Riesz representation theorem for each $Z = z$ and obtains an objective function which is
not in the form usually seen in regression [G, eq. (5)]. Although this is equivalent to our squared-loss, and therefore
leads to the same empirical estimates, our approach facilitates the use of other loss functions more easily, as well as
more complex kernels, which will lead to different empirical estimates; please see **Reviewer #2, 2, 3**. (ii) We believe
our theoretical analysis is more thorough than [G], as we derive a new result for convergence rate and provide new
analysis of what the surrogate loss means exactly (L266-L286), which we believe is more complete than [G, Thm 3.2].

**[5.3]** This is a valid point; we agree that the paper would read better if we move (at least parts of) Appendix B into the
main body of the paper. This was purely due to space constraints. We propose to do so in the camera-ready version.
**[5.4]** This is a good point – we propose to mention the uncentred case in the camera-ready version, since they are often
used in the literature. The advantages of our CME approach remain true against the uncentred case.

**Reviewer #2** We thank the reviewer for the overall positive view. For the first concern, please see **Reviewer #1, [3.4]**.
**2, 3.** This is a great comment. It is true that, despite its simplicity and intuitiveness, the squared-loss does come
with some disadvantages, particularly in terms of robustness. By using a different loss on L219, our new formulation
facilitates the use of other loss functions conveniently for a more robust RKHS representation of conditional distributions,
and we believe [Laforgue et al., 2019] would be a relevant paper to cite. Moreover, on L221, it is not necessary to
use a scalar kernel composed with identity; any other operator-valued kernel can be used, and different empirical
estimates will be obtained than the closed-form in (3). This ability to facilitate other loss functions and OVKs is a
definite advantage of our new formulation over the existing approach, and we propose to emphasise this point in the
camera-ready version. The subsequent closed-form empirical estimates and theoretical analysis that follow are based on
the squared-loss and scalar kernel composed with identity, and we propose to keep these as they are, since this case is
the most basic and common case and we believe the results are therefore still of value.
**4.** This is a great point, and we have actually been working precisely in this direction, including developing this into a
statistical test. Expectation over $Z$ would be a good way of aggregating the random criterion into a single number, and
indeed we can also apply it to HSCIC, but there are also other ways of doing it, e.g., considering the vv-RKHS norm of
the difference of the estimations. We argue that this is an example of how our new formulation of CMEs can open up
new questions and applications. In this paper, we left it random to retain the direct analogy with the unconditional case.
**5.** As long as the conditioning variables are absolutely continuous with respect to each other, they are not required to be
the same. We propose to let the conditioning variables be different in the camera-ready version, for full generality.

**Reviewer #3** We thank the reviewer for a critical review. For the main concern over practicality, we believe the shift of
formulation we propose is such that it enables new applications that were not previously possible. A concrete example
is laid out in **Reviewer #2, 4**, and a real-world application could be the estimation of the distributional treatment
effect, comparing P(Y|X,T=0) and P(Y|X,T=1) where Y, X, and T denote outcome, covariate, and treatment variables,
respectively. This is an important problem in medicine, public policy, and economics. The CME was first introduced in
the machine learning community (Song et al., ICML2009) and most important papers in this area have been published
in ICML, NeurIPS, AISTATS, etc. Hence, we believe NeurIPS is the right venue (see also **Reviewer #1, [3.1]**). Also,
we will further improve clarity in our camera-ready version (in particular, please see **Reviewer #1, [5.3]**).

**Reviewer #4** We thank the reviewer for the positive view of our paper. As for your concern, unfortunately, the previous
stronger assumptions are virtually always violated. For the relevant discussion and references, please see L131-147.

[Meta-Review · NeurIPS 2020]

Three referees support acceptance for the contributions, finding the novel theoretical formulation of the CME to be a valuable contribution to the community. Referee R3 indicated rejection; the rebuttal disputes R3's claim that the theoretical contribution is not sufficient. I discount R3 because no concrete reason was given for their assertion that the contribution is insufficient. I agree with the remaining reviewers that the contribution is valuable and of interest to the NeurIPS community. The authors promised several changes to be made in the camera ready version to clarify the results; please do implement these changes.